

# Decadal changes in phytoplankton functional composition in the Eastern English Channel: evidence of upcoming major effects of climate change?

Zéline Hubert[1], Arnaud Louchart[1,2], Kévin Robache[1], Alexandre Epinoux[1,3], Clémentine Gallot[1,4], Vincent Cornille[1,5], Muriel Crouvoisier[1], Sébastien Monchy[1], and Luis Felipe Artigas[1]

[1]Université Littoral Côte d'Opale, Université de Lille, CNRS, IRD, UMR 8187 LOG, Laboratoire d'Océanologie et de Géosciences, F62930, Wimereux, France
[2]Netherlands Institute of Ecology (NIOO-KNAW), Department of Aquatic Ecology, Droevendaalsesteeg 10, 6708 PB Wageningen, The Netherlands
[3]Université Brest, Ifremer, CNRS, IRD, LOPS, F-29280 Plouzané, France
[4]Mediterranean Institute of Oceanography (MIO), Campus de Luminy, 163 Av. de Luminy, 13288 Marseille cedex 9, France
[5]Ifremer, Unité Littoral, Laboratoire Environnement et Ressources, 150 quai Gambetta, 62321 Boulogne-sur-Mer, France

**Correspondence:** Zéline Hubert (zelior effne.hubert@univ-littoral.fr) and Luis Felipe Artigas (felipe.artigas@univ-littoral.fr)

**Abstract.** Global change is known to exert a considerable impact on marine and coastal ecosystems, affecting various parameters such as sea surface temperature, rain-off, circulation patterns, and the availability of limiting nutrients like nitrogen, phosphorus and silicon, each influencing phytoplankton communities differently. This study is based on weekly to fortnightly *in vivo* phytoplankton observations in the French waters of the Eastern English Channel at fine spatial resolution ($\sim$ 1 km)

along an inshore-offshore gradient in the Strait of Dover. Phytoplankton functional composition was addressed by automated 'pulse shape-recording' flow cytometry, coupled with analysis of environmental variables over the last decade (2012-2022). This method allows for the characterization of almost the entire phytoplankton size range (from 0.1 $\mu$m to 800 $\mu$m width) and the determination of the abundance of functional groups based on optical single-cell signals (fluorescence and scatter). We explored seasonal, spatial, and decadal dynamics in an environment strongly influenced by tides and currents. Over the past

11 years, sea surface temperatures showed an increasing trend in all stations, with nearshore waters warming faster than offshore waters (+1.063 °C vs. +0.929 °C). Changes in nutrient concentrations have led to imbalances in nutrient ratios (N:P:Si) compared to Redfield molar reference ratios, though a rollback (2012-2018) to balanced ratios (since 2019). Phytoplankton total abundance has also increased over the decade, with a higher contribution of small-size cells (picoeukaryotes and picocyanobacteria) and a decrease in microphytoplankton, particularly near the coast. The winters of 2013-2014 and 2019-2020

have been identified as shifting periods in this time series. This study provides the first assessment of decadal changes of the whole phytoplankton community by an automated *in vivo* single-cell approach, which will need to be explored further in the frame of changes in trophic transfers and water quality.





# 1   Introduction

As the main primary producer of marine ecosystems, marine phytoplankton play a crucial role in structuring pelagic food
webs and greatly influence biogeochemical cycles in the ocean. This polyphyletic group exhibits a wide range of sizes (from
less than a micron to centimetres), shapes, single-cell or colonial forms, life stages, pigments, storage products, motility, re-
productive rates and more (Simon et al., 2009). All these functional traits, especially size, will determine their involvement
and performance in biogeochemical cycling (e.g. carbon fixation, nutrient uptake; Hillebrand et al., 2022), their growth rate
(Marañón, 2015) as well as energy transfer efficiency in higher food webs (Mehner et al., 2018). The abundance, the com-
munity composition and the succession of different phytoplankton groups are rapidly regulated by environmental parameters
(temperature, light availability, nutrient disponibility) and biotic interactions (Margalef, 1978; Winder and Sommer, 2012;
Barton et al., 2013; Rombouts et al., 2019). Due to the rapid turnover between generations and response of communities to
environmental changes, phytoplankton is used as an indicator to assess the ecological status of pelagic marine ecosystems.
In the context of the Marine Strategy Framework Directive (MSFD, 2008/56/EC), phytoplankton diversity, composition and
abundance are used to assess ecological status of pelagic habitats (Louchart et al., 2023a, b; Holland et al., 2023a) and to study
marine eutrophication (Rombouts et al., 2019).

In addition to local pressures, climate change significantly influences environmental parameters in marine systems, leading
to rising temperatures, changes in light intensity, rainfall and river flow (Cooley et al., 2022). Coastal and shallow environments
are particularly vulnerable to these changes (Cloern et al., 2016). While these global-scale modifications are already observed
at regional levels, they have not yet been observed at the sub-mesoscale (Capuzzo et al., 2018). The Eastern English Channel
(EEC) is a shallow marginal sea under a macrotidal regime and heavily influenced by human activity. It is an exploited ecosys-
tem for fisheries, hosting major harbours such as Cherbourg, Le Havre, Boulogne-sur-Mer and Calais for the French coast.
The EEC is subjected to an intense maritime traffic, particularly around the Strait of Pas-de-Calais - Dover, connecting the
English Channel to the North Sea, which ranks as the world's second busiest strait. Furthermore, the coastline is largely cov-
ered by agricultural land, leading to potential nutrient and/or pesticide inputs into coastal waters through rainfall. In addition,
the EEC coast is characterized by numerous estuaries, including the Seine, the Somme and smaller estuaries (Authie, Canche,
Liane, Wimereux and Slack) until the Strait of Dover, which collectively contribute to the 'coastal flow' generating significant
terrigenous inputs (Brylinski et al., 1991). Over the last 150 years, the English Channel has witnessed a rise in precipitation
(Scholz et al., 2022) and a notable increase in temperature since the 1990s was observed in its Eastern part (McLean et al.,
2019; Tinker et al., 2020). On the other hand, changes in nutrient concentrations were observed after implementation of the
European Common Agricultural Policy (CAP), resulting into stronger phosphorus mitigation effort than nitrogen, leading to
an imbalanced N:P ratio (Loebl et al., 2009; Talarmin et al., 2016; Lheureux et al., 2023). These modifications are expected to
have consequences on phytoplankton communities in the EEC, affecting their abundance, composition, size, and bloom timing
(Falkowski and Oliver, 2007; Sommer and Lengfellner, 2008; Winder and Sommer, 2012; Henson et al., 2018; Rombouts et al.,
50  2019).



Previous time-series studies have revealed a decline in phytoplankton biomass in the EEC using chlorophyll *a* measurements (Lefebvre et al., 2011; Gohin et al., 2019). Over the past decade, analysis of Continuous Plankton Recorder (CPR) data has shown a change in phytoplankton composition in the North Sea, characterized by an increase in small diatoms and dinoflagellates (Holland et al., 2023b). In the EEC, the years between 1992 and 2007 could be categorized according to the dominance of the haptophyte *Phaeocystis globosa* or diatoms (Lefebvre et al., 2011). Previous long-term studies in the EEC, based on satellite images, chlorophyll *a*, microscopy or CPR, enable to adress some changes in phytoplankton phenology and diversity, but they neglect picophytoplankton and some nanophytoplankton. In the Western English Channel, it was shown that small phytoplankton represented 99.98% and 71% of phytoplankton abundance and biomass (McQuatters-Gollop et al., 2024) and that temperature increase can lead to changes in the structure and cell size of this community (Zohary et al., 2021). However, these compartments were overlook by microscope observations, while playing a significant role in marine ecosystems by recycling nutrients and dissolved organic matter(microbial loop), and the export of carbon to higher trophic level through zooplankton consumption. In a context of climate change, a modification in the balance between pico- and nanophytoplankton communities structure could increase the importance of microbial loop and microbial food webs, reduce carbon sequestration (respiration, carbon fixation, ocean carbon export), change the trophic pathways and *in fine* influence higher trophic levels including fisheries (Falkowski et al., 2000; Laws et al., 2000; Hillebrand et al., 2022).

In this study, we used data acquired regularly since 2012 on the full size range of phytoplankton, including picophytoplankton, addressed *in vivo* by automated 'pulse shape-recording' flow cytometry, coupled with environmental variables. Some previous studies applying this approach in the EEC were performed describing seasonal (Bonato et al., 2016) and short inter-annual changes (Breton et al., 2017), as well as spatial and temporal variability on oceanographic cruises (Bonato et al., 2015; Louchart et al., 2020, 2024). The aim of this study was to identify and quantify changes at sub-mesoscale and to report for the first time decadal trends on the entire phytoplankton community. The approach combined high frequency compared to most reference observation networks, and high spatial resolution across all water types, from nearshore to offshore waters, in a frontal system by the Strait of Dover. To characterize these trends and assess magnitude of phytoplankton communities change over a decade, we applied a functional community composition approach. This approach considered temporal changes in biomass, abundance and composition, relative to changes in environmental variables, from single-cell up to community levels.

## 2 Materials and Methods

### 2.1 Study area and sampling strategy

Subsurface marine samples were collected weekly to fortnightly, from February 2012 to December 2022, on board the *Sepia II* (CNRS INSU-FOF) R/V. Sampling was conducted along a nearshore-offshore transect located by the Dover Strait (EEC), known as DYPHYRAD monitoring. This transect consists of 9 sampling stations (Fig. 1), from R0 (50°8' N; 1°59' E) to R4 (50°8' N; 1°45'22 E), spaced between 0.8 kilometers to 1.7 kilometers. Stations characterized three zones (from offshore to nearshore): offshore (R4, R3, R3'), frontal (R2, R1') and nearshore (R1, R0', R0), in order to facilitate the description of spatial phenomena, according to Brylinski et al. (1991).



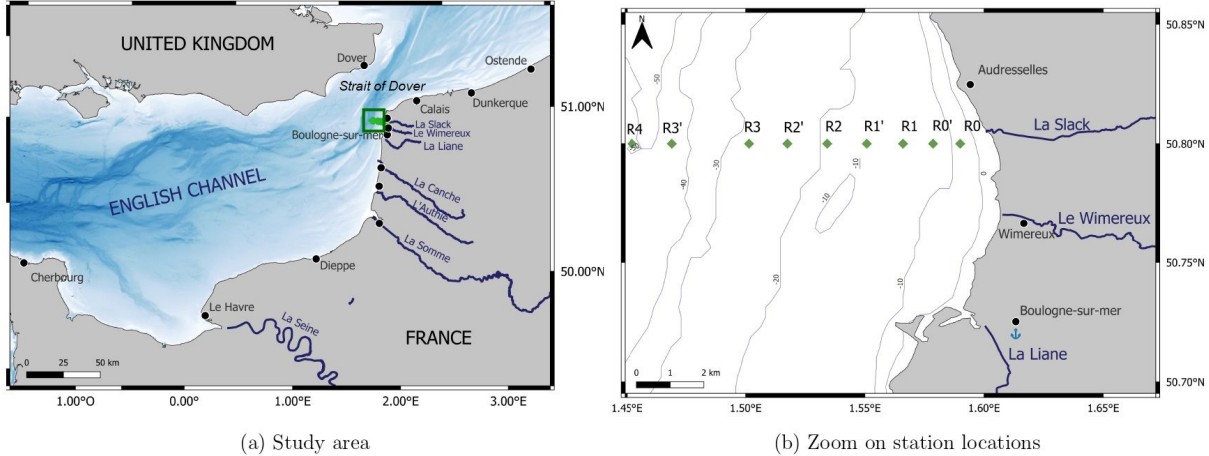

(a) Study area

(b) Zoom on station locations

**Figure 1.** Map of the study area (a) the Eastern English Channel and (b) location of DYPHYRAD stations off the Slack River estuary by the Strait of Dover.

## 2.2 Environmental parameters

Temperature (SST, °C) and salinity (S, PSU) were recorded at each sampling station sub-surface (1 to 2 meters depth) with a Conductivity Temperature Depth (CTD) probe (SBE19 + and SBE 25, SeaBird Ltd, United States). Subsurface water layer (1 meter depth) were collected using a Niskin bottle. Dissolved inorganic nutrient concentrations ($NO_2^-$, $NO_3^-$, $SiO_2$ and $H_3PO_4$) were measured at the main sampling points (R0, R1, R2, R3, R4). Seawater samples were collected and kept cooled in the dark until return to the laboratory where it was frozen (-20 °C) until analysis. Nutrient concentrations were obtained

using an autoanalyser (AutoAnalyzer ALLIANCE SpA, Italy and, since 2016, a AA3 HR AutoAnalyzer, SEAL Analytical GmBH, Germany), following the French coastal observation network 'Service d'Observation en Milieu Littoral' (SOMLIT) protocol (Garcia and Oriol, 2019; Breton et al., 2023). Nitrogen will be referred to as the sum of nitrite ($NO_2^-$) and nitrate ($NO_3^-$).

## 2.3 Phytoplankton biomass, abundance and size

Phytoplankton biomass was estimated through chlorophyll *a* concentration analysis. Between 250 mL to 1 L of seawater were filtered on 47 mm diameter GF/F (Whatman) filters and then stored at -80 °C until pigment analysis, after extraction on 90 % acetone at 4 °C overnight. Chlorophyll *a* and degraded pigments (phaeopigments) concentrations were measured both before and after acidification (HCl 0.2 mol L$^{-1}$) on a Turner Designs benchtop fluorometer (10-AU Field Fluorometer, Turner Designs Ltd, USA) following the protocol developed by Holm-Hansen et al. (1965) and the equations developed by Lorenzen (1967).

Phytoplankton functional composition was obtained from *in vivo* samples using CytoSenses (Cytobuoy b.v., Netherlands). The flow cytometers are equipped with a blue laser (488 nm, 50 mW) to allow discrimination between phototrophic and non-phototrophic particles. Flow cytometers provide counting of particle for sizes from 0.1 to 800 $\mu$m width, to consider practically





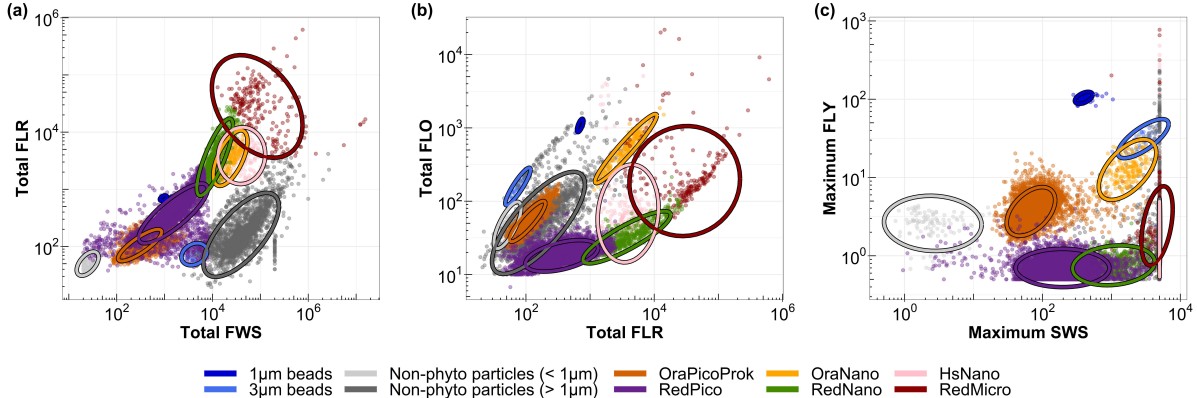

**Figure 2.** Cytograms of EEC used to characterize the main phytoplankton groups: (a) total red fluorescence vs total forward scatter for the discrimination of picoeucaryotes (RedPico), nanoeucaryotes (RedNano) and microeucaryotes (Red Micro), (b) total red fluorescence vs total orange fluorescence for the discrimination of *Synechococcus* spp. (OraPicoProk) and Cryptophytes (OraNano), (c) maximum yellow fluorescence vs maximum sideward scatter for the discrimination of *Coccolithophoridaea* (HsNano). Ellipses on the graphs are calculated from a *t*-distribution at 95 % confidence level, aiding in accurate delineation of the respective phytoplankton groups.

the whole phytoplankton size-range. Technical specifications for the flow cytometer can be found in previous studies using this instrument at Laboratory of Oceanology and Geosciences (LOG; Bonato et al., 2015, 2016; Louchart et al., 2024). Each

sample underwent analysis using two separate protocols, each targeting specific size and optical parameters. The first protocol, referred to as the 'Pico' protocol, used a low detection threshold (around 10 mV red fluorescence), a low pump speed (5 $\mu$L s$^{-1}$) and a short sampling time (5 minutes). This protocol targeted cells ranging from 0.1 to 3 $\mu$m in size, characterized by low fluorescence and high abundance. The second protocol focused on nano– and microphytoplankton, using a higher detection threshold (around 25 mV red fluorescence), a high pump speed (between 10 and 13 $\mu$L s$^{-1}$) and a long sampling time (8 to

10 minutes).

Manual discrimination and characterization of six main Phytoplankton Functional Groups (PFGs) was performed on the basis of their size distribution, structure complexity and fluorescence signals, in accordance with the interoperable vocabulary of Thyssen et al. (2022). Cytogram analysis (biplot combining scatters or fluorescence) was performed using CytoClus 4 software (Cytobuoy b.v., Netherlands). Several of these functional groups have been previously identified in the area, includ-

ing OraPicoProk (e.g. *Synechococcus* spp. type cells), RedPico (e.g. picophytoplankton), RedNano (e.g. nanophytoplankton, mainly dominated by *Phaeocystis globosa* during the spring bloom; Bonato et al., 2015, 2016; Guiselin, 2010), HsNano (e.g. coccolithophore type cells), OraNano (e.g. cryptophyte type cells), and RedMicro for microphytoplankton (Fig. 2).

For the final PFG dataset, only picoeukaryotes (RedPico) and cyanobacteria (OraPicoProk) were considered in the 'Pico' protocol. The other groups (RedNano, OraNano, HsNano and RedMicro) were classified using the 'Micro' protocol. To accu-

rately perform phytoplankton functional group discrimination and labeling, we used 1 and 3 $\mu$m fluorescent beads (labelled with yellow and multi-fluorescence dyes, respectively).





## 2.4 Statistical analysis

All data analysis, graphical representations and statistical analyses were carried out using R software (R-project, CRAN) version 4.3.1. The plots were produced using the 'ggplot2' package, version 3.5.0. Date management was implemented using the 'lubridate' package, version 1.9.3. Multivariate statistical analyses were performed using the 'vegan' package version 2.6-4 and trend tests were performed using the 'trend' package version 1.1.6.

### 2.4.1 Spatial and seasonal pattern

Spatial and annual variability of environmental parameters and phytoplankton communities were investigated along the DY-PHYRAD transect. Stations were not uniformly sampled due to difficult weather conditions. Thus, we applied a linear time series interpolation on this station to fill these gaps and define regular and complete sampling intervals. The abundance data was $\log_{10} +1$ transformed in order to reduce the weight of high abundance in the analyses. Seasonal dynamics was evaluated by applying a Generalized Additive Model (GAM). This statistical model develops linear regression by considering non-linear relationships between dependent and independent variables through the use of smooth functions. In this study, GAM facilitated the exploration of variability in phytoplankton functional group abundance over time using smooth spline estimation, as shown in the following formula:

$$g(abundance) = S_0 + S(year) + \epsilon, \epsilon\ N(0, \sigma^2) \tag{1}$$

where $S_0$ is the intercept, S the smoothing function, $\epsilon$ the GAM regression and $\sigma$ the standard deviation.

This method facilitates the modelling of non-linear relationships between the time factor and the abundance variable. We applied these GAMs individually to each PFG within every station.

### 2.4.2 Spatio-temporal interaction

To assess the spatio-temporal variability of PFGs over the different years, seasons and stations included in the time series, we used PERmutational Multivariate ANalysis Of VAriance (PERMANOVA). This statistical method is particularly robust because it is non-parametric and relies on permutations in the context of the Bray-Curtis distance matrix that was used. Before conducting the analysis, we standardized abundance values using the Hellinger transformation as proposed by Legendre and Gallagher (2001) to reduce the influence of the most dominant groups while preserving the contribution of rare groups. This standardisation is often applied to abundance data, as it preserves the proportions between groups while reducing the effect of extreme values maintaining the distances between samples. The strength of PERMANOVA lies in its permutation-based testing approach, making it resilient against assumptions about data distribution. In this study, we performed 999 permutations to ensure robustness and statistical validity of the results. In the event of a significant difference within a parameter (year, season, station), a post-hoc Tukey multiple comparison test (or Tukey's Honestly Significant Difference) was performed (Tukey, 1949) to determine which of all possible pairs have a significant difference at a 95 % confidence interval.





### 2.4.3 Decadal changes and trends

The analysis of decadal changes and trends in time series was based on the processing of raw data, which were averaged on a monthly basis to establish a consistent and regular time interval. To analyze changes without eliminating the seasonal cycle, we have subtracted the monthly average for each year from the monthly average for the entire period under consideration.

The cumulative sums method was used to analyse trends and patterns of the time series dataset after checking the non-normality of the data with the Shapiro-Wilk test (Shapiro and Wilk, 1965). This method is particularly robust in the case of data series with gaps, noise or following a non-normal distribution (Regier et al., 2019). The cumulative sums corresponded to the successive addition of each anomaly value in a chronological order. By analysing these cumulative sums over time, we were able to define periods of below-average values (in the case of a decreasing slope) and above-average values (in the opposite case of an increasing slope; Regier et al., 2019) and deduce phases of increasing or decreasing parameters of interest. Moreover, a change in the direction of the slope can be used to identify inflexion points in the series (Regier et al., 2019).

The Mann-Kendall trend test was applied to determine the general direction of trends over time (monotonic trend; Mann, 1945; Kendall, 1948) to obtain the general sign of the slope (by adding together all the signs two by two) for each parameter. The Mann-Kendall test gives no indication of the magnitude of the trend but only its sign. This was combined with a Sen slope calculation to quantify the magnitude of change within the series (Sen, 1968). This non-parametric test was used to obtain a slope value corresponding to the median of all slopes (expressed in units per decade) in the series (in pairs). These trend tests were carried out on the whole series, analyzing trends by station in order to observe small-scale spatial variations.

### 2.4.4 Nutrients imbalance

Potential nutrient limitations over the last decade were identified through a diagram of Si:N:P molar ratios wheere data were averaged by year. This diagram is based on the ratios Si:N = 1:1, N:P = 16:1 and Si:P = 16:1 previously described by Redfield et al. (1963) and Brzezinski (1985). To improve visualization, the axes were transformed into $\log_{10}$ and the graph was divided into six zones, each describing a nutrient limitation.

## 3 Results

### 3.1 Seasonal pattern along a nearshore-offshore gradient

The Generalized Additive Model (GAM) modelling environmental parameters, offered valuable insights into the transect dynamic characteristics throughout a standard year (Fig. 3). Temperatures peaked between the day 220 and day 240 (Julian day), depending on the station, preceded by minimum values between the day 30 and day 60 of the year. Spatial fluctuations in sea surface temperature were nuanced, with slight shifts between increases and decreases. In particular, nearshore waters showed greater reactivity than offshore waters, with temperature colder in winter and warmer in summer, earlier than in offshore waters. Salinity was highest in summer, showing a gradual increase from winter values and an abrupt decrease in the summer-fall transition. Salinity revealed a distinct and contrasting spatial pattern between nearshore and offshore waters, with station R0



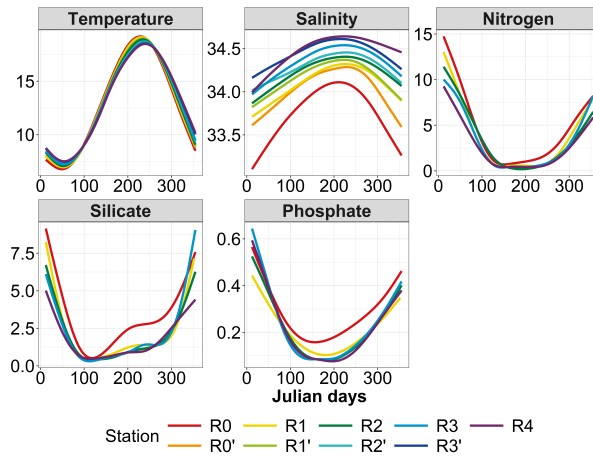

**Figure 3.** Seasonal and spatial variability of environmental parameters: temperature (°C), salinity (PSU), nitrogen ($\mu$mol L$^{-1}$), silicate ($\mu$mol L$^{-1}$) and phosphate ($\mu$mol L$^{-1}$). The smoothed curves are modelled per Julian day using GAM.

consistently displaying lower salinity levels compared to others stations along the transect. Salinity levels increased progressively from coastal towards offshore waters, punctuated by intermittent periods of inversion, such as those observed between day 1 and day 70 (March 11$^{th}$) at station R4, where salinity fell below that of R3'. The spatial difference was less marked in summer compared to winter. Nutrient concentrations also showed a seasonal pattern, starting with high values during the first months of the year (January-February), followed by a decline in spring, before increasing again from summer to autumn-winter. Silicate showed a sharp depletion from offshore to nearshore waters with lower concentrations around day 110, with a notable early increase observed at station R0 around day 200 (July, 19$^{th}$) compared to a later increase in the other stations. Phosphate and nitrogen concentrations showed similar temporal dynamics, both declining in spring, later than silicate concentration, following different trends on nearshore stations. Nitrogen concentration increased from offshore to nearshore areas, and was almost depleted in late spring and summer, whereas R0 showed an slightly earlier summer-fall increase compared to offshore waters. Phosphate showed a more complex pattern, with higher phosphate concentration in nearshore station R0 in spring and an increase from day 155, earlier than the rest of the stations (increase registered from day 180 to 220).

Chlorophyll *a* concentration showed a pronounced increase early in the year reaching higher values (spring bloom) from day 85 to 95 in all stations and more particularly at station R0', followed by a decrease to values similar to stations R0, R1 and R1' (Fig. 4, left). A strong spatial gradient was evidenced grouping the first 4 nearshore stations, the two frontal stations (R2 and R2') and the gradient on offshore stations (R3, R3', R4). It was much more pronounced during the bloom period than during the rest of the year with the most littoral station (R0) concentration decreasing to offshore levels from late spring. On the other hand, total abundance showed a pattern opposite to that of chlorophyll *a*, with a minimum abundance in spring and a maximum in summer (Fig. 4, right). An increase in total abundance was evidenced from spring only in the two nearshore stations R0 and R0', whereas a marked spatial pattern was observed from late spring-summer to early fall, with decreasing



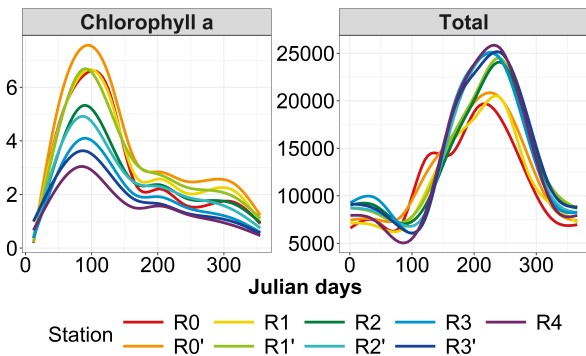

**Figure 4.** Seasonal and spatial variability of total phytoplankton parameters: chlorophyll $a$ ($\mu$g L$^{-1}$) and total abundance (cells mL$^{-1}$). The smoothed curves are modelled per Julian day using GAM.

abundance from offshore waters (R4, R3, R3'), to frontal area (R2', R2, R1'), reaching the lowest cell abundance in nearshore waters (R1, R0', R0).

The GAM analysis revealed a relatively high variability across space and over time, for the six PFGs (Fig. 5). The seasonal heterogeneity was most striking across the PFGs, rather than across different water bodies, considering a single PFG. However, RedMicro and HsNano (and, to a lesser extent, RedNano in spring) presented a marked spatial heterogeneity. PFGs with phycobilins dominance (OraPicoProk and OraNano) reached their lowest abundance in spring (April-May) and their highest abundance during summer- early fall period (July-September). The abundance of these PFGs increased along the nearshore-offshore transect. On the other hand, the abundance of PFGs with chlorophyll $a$ dominance (RedPico, RedNano, RedMicro and HsNano) decreased along the neashore-offshore transect. The seasonal pattern of RedPico followed those of OraPicoProk and OraNano. The seasonality of RedNano was characterised by a highest abundance in spring (April-May) and a lowest in summer with minimum values in autumn-winter. Throughout the rise and fall of the spring bloom, the RedNano group displayed almost no apparent spatiality dynamics, albeit some spatial difference considering their total abundances. During the autumn-winter period, the nearshore-offshore pattern of RedNano disappears, replaced by a different spatialization with a higher abundance in the middle of the transect, and lower abundances at the extreme stations (R0 and R4). RedMicro abundance increased from January to July before dropping. Towards the end of the year (from day 250), RedMicro abundance was the highest in stations R0' and R1'. HsNano abundance was more variable than that of any other PFG, maybe because of low occurrence of high scattering PFG in the study area (coccolithophores and thecate dinoflagellates).

## 3.2 Spatial and temporal interaction in the Dover Strait dynamics

Over the last decade, environmental variables and phytoplankton communities significantly explained 45 % and 39 % of the respective variances (PERMANOVA p-value < 0.05) with a strong difference according to the F-statistic score (Table 1, and 2). The sampling year was the most important factor in explaining the variance and influence on phytoplankton abundance





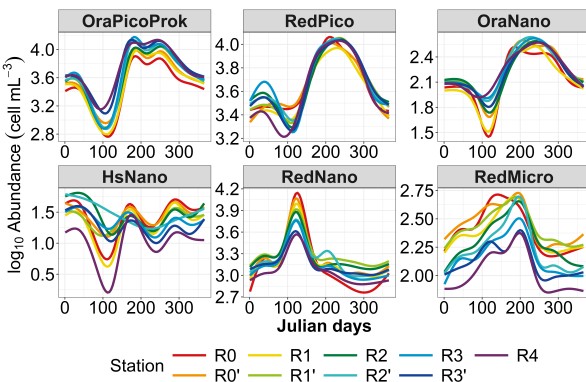

**Figure 5.** Seasonal and spatial variability of Phytoplankton Functional Groups abundance. The smoothed curves are modelled per Julian day using GAM.

(15 %) and on environmental variables (11 %). Finally, over the whole decade, stations location along the transect (expressed
in Longitude) only explained 5 % very little of the variances for phytoplankton abundance and environmental variables. The combined factors of year and season explained 6.9 % of the variance for phytoplankton abundance groups and 9.8 % of the variance for abiotics parameters. Pairwise post-hoc tests showed that all seasons differed significantly (p-value $< 0.05$) from one another in terms of abiotic parameters, with the exception of autumn and winter for phytoplankton communities. No significant differences were observed between stations regarding phytoplankton communities, with the exception of R0 that
was significantly different from all stations for environmental parameters according to Tukey's post-hoc test (p-value $< 0.05$). Considering the interannual variability, 2020 differed significantly from all other years (except 2015) for abiotic parameters, while for phytoplankton communities it differed from 2012 and 2017. In addition, abiotic parameters for 2015 and 2022 also differed from other years in the series (2012, 2013, 2017, 2021, 2022 and 2014, 2015, 2019, 2020 respectively). Phytoplankton communities are particularly different from 2017 onwards, with 2020 and 2021 being the most different from all other years.

## 3.3 Long-term variability

### 3.3.1 Environmental decadal evolution

Between 2012 and 2022, the sea surface waters within nearshore-offshore transect exhibited notable fluctuations in temperature, salinity, and concentration of key nutrients such as nitrite, nitrate, phosphate and silicate. These variations were analyzed as part of a global approach incorporating both cumulative sums and general trends for each parameter over time. At the
240 beginning of the time series, the cumulative sum analysis for temperature (Fig. 6a) indicated below-average values (negative slope) influenced by starting value. Since 2014, an overall trend of temperatures increase was observed with consistently above-average temperatures (positive slope). Besides some fluctuations, raw data and decadal temperature trend analysis corroborated this observation, revealing an overall increase ranging from +0.89 to +1.21 °C between February 2012 and December 2022 (Sen





**Table 1.** PERMANOVA partitioning and analysis of environmental variables (temperature, salinity, nitrogen, phosphate, silicate) from the decadal data, based on range-transformed values and Bray–Curtis dissimilarities. Df stands for degrees of freedom, the coefficients of determination ($R^2$) explaining the variability of the dependent variable. The F statistic evaluates the size effect, the higher the F, the greater the variation. Bold indicates a significant effect on variability (p-value $< 0.05$).

| Source | df | $R^2$ | F-statistic | p-value |
|---|---|---|---|---|
| Year | 10 | 0.114 | 26.39 | **0.001** |
| Station | 4 | 0.050 | 28.89 | **0.001** |
| Season | 3 | 0.448 | 344.70 | **0.001** |
| Year × Station | 40 | 0.01 | 0.626 | 0.995 |
| Year × Season | 26 | 0.098 | 8.72 | **0.001** |
| Station × Season | 12 | -0.002 | -0.44 | 1.000 |
| Year × Station × Season | 95 | 0.021 | 0.51 | 1.000 |

**Table 2.** PERMANOVA partitioning and analysis of phytoplankton abundance from the decadal data, based on Hellinger-transformed abundances and Bray–Curtis dissimilarities. Df stands for degrees of freedom, the coefficients of determination ($R^2$) explaining the variability of the dependent variable. The F statistic evaluates the size effect, the higher the F, the greater the variation. Bold indicates a significant effect on variability (p-value $< 0.05$)

| Source | df | $R^2$ | F-statistic | p-value |
|---|---|---|---|---|
| Year | 10 | 0.150 | 50.48 | **0.001** |
| Station | 8 | 0.011 | 4.58 | **0.001** |
| Season | 3 | 0.391 | 437.16 | **0.001** |
| Year × Station | 80 | 0.009 | 0.38 | 1.000 |
| Year × Season | 27 | 0.070 | 8.66 | **0.001** |
| Station × Season | 24 | 0.003 | 0.42 | 1.000 |
| Year × Station × Season | 194 | 0.017 | 0.29 | 1.000 |

slope values, p-value $< 0.05$). Nearshore waters showed more pronounced warming compared to offshore waters (Table 3).
Sea surface salinity (Fig. 6b) began with a phase of decline until winter 2013, increases until winter 2019 then declines again until the end of the studied period. Although trend tests failed to detect any significant trends in salinity values over the period (Table 3), all values were negative and, in line with the fluctuations observed in both raw data and cumulative sums. Regarding nutrient levels, nitrogen concentrations (Fig. 6c) displayed a 'U' shape pattern throughout the decadal period, decreasing from particularly high values during winter 2013-2014 (N0x $> 20$ $\mu$mol L$^{-1}$) and then increase to high values from January 2018 and in winter 2021-2022 (N0x $> 10$ $\mu$mol L$^{-1}$). However, a significant decrease (Mann-Kendall trend analysis) in nitrogen was observed at stations R0 and R1 (nearshore waters) during the whole period (Table 3). The cumulative




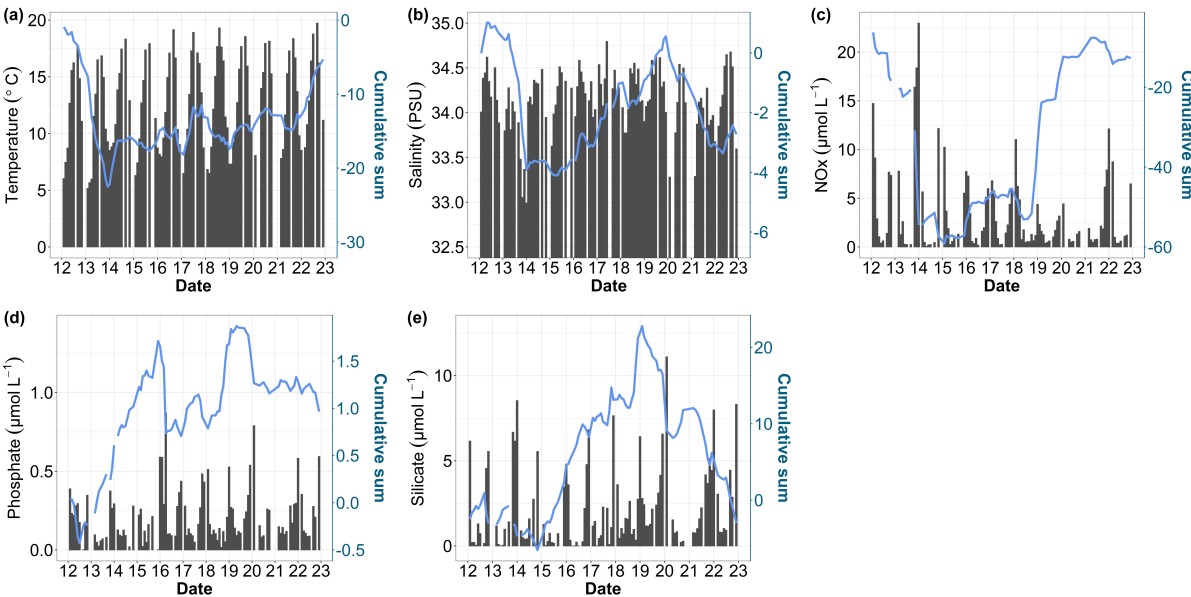

**Figure 6.** Time series of environmental parameters: (a) temperature, (b) salinity, (c) nitrogen, (d) phosphate and (e) silicate. Bar plot represents monthly raw data (left y-axis). The blue lines correspond to the cumulative sum of anomalies over time (right y-axis).

**Table 3.** Trends and magnitude of change of temperature, salinity, nitrogen ($NO_2 + NO_3$), phosphate and silicates. Bold indicates a significant trend (p-value < 0.05) over the period 2012-2022. The figures indicate the magnitude of the trend.

| Parameters (Units) | R0 | R0' | R1 | R1' | R2 | R2' | R3 | R3' | R4 |
|---|---|---|---|---|---|---|---|---|---|
| Temperature (°C) | **+1.05** | **+1.18** | **+1.21** | **+1.01** | **+0.94** | **+0.89** | **+0.95** | +0.95 | **+0.93** |
| Salinity (psu) | -0.15 | +0.07 | -0.035 | -0.05 | -0.05 | -0.099 | -0.10 | -0.12 | -0.07 |
| Nitrogen ($\mu$mol L$^{-1}$) | **-1.025** | - | **-0.94** | - | -0.29 | - | -0.264 | - | 0.127 |
| Phosphate ($\mu$mol L$^{-1}$) | **+0.09** | - | **+0.05** | - | +0.025 | - | **0.000** | - | **-0.026** |
| Silicate ($\mu$mol L$^{-1}$) | +1.21 | - | +0.75 | - | **+1.08** | - | **+1.38** | - | **+1.39** |

sum analysis of phosphate concentrations revealed a more intricate pattern (Fig. 6d), characterized by alternating phases of increase and decrease, punctuated by peaks in winter 2015-2016 and 2019-2020. Trend analysis indicated an overall increase in phosphate concentration at nearshore stations (R0 and R1, Table 3). Cumulative sums of silicate concentration depicted a
declining trend since winter 2013-2014 except during winter 2019-2020 (Fig. 6e). Raw data highlight elevated concentrations during the winter of 2019-2020. Significant increases in silicate levels were detected from R2 (frontal waters) to R4 (offshore waters) stations (Table 3). The combined analysis of raw data, cumulative sums and trend tests facilitated the identification of trends and periods of change in physico-chemical variables, such as the transition between 2013 and 2014, as well as the period from 2018 to 2020.



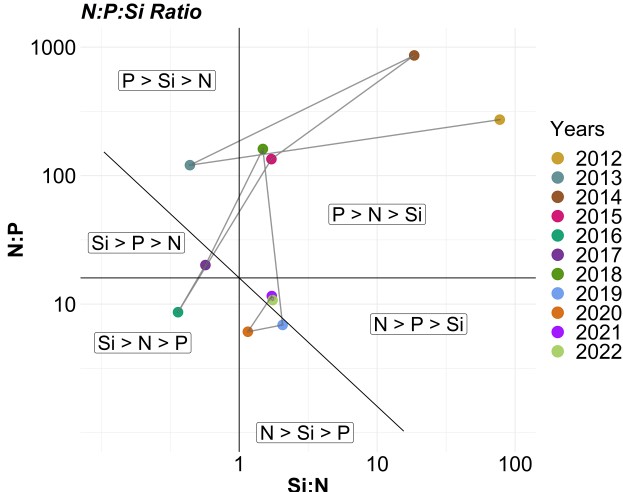

**Figure 7.** Evolution over time of potential N:P:Si nutrient limitations according to the Redfield molar ratio (C:N:Si:P = 106:16:16:1; Redfield et al., 1963). The horizontal black line represents the N:P limit of 16:1, the vertical black line the Si:N ratio of 1:1 and the diagonal black line the Si:P = 16:1 ratio. The colors correspond to the average ratio for each year.

Fluctuations in nutrient concentration have implications on Redfield ratios and in turn underline potential nutrient limitation. Consequently, phytoplankton would respond to these variations through changes in the community composition, biomass and productivity. The assessment of interannual averages across all stations was conducted to investigate annual nutrient potential limitations (Fig. 7). From 2012 to 2015, the system exhibited indications of potential phosphate limitation (Fig. 7, top part), corresponding to the elevated nitrogen values observed at the beginning of our time series (Fig. 6c). In 2016 and 2017, the Redfield ratios shifted towards a potential silicate-limited system (Fig. 7, bottom left). Since 2019, a trend towards a potential nitrogen limitation becomes apparent. However, since 2020, the system seemed to be moving towards an equilibrium in the N:P:Si ratio. These shift periods aligned with the breakpoints identified previously with the cumulative sums of nitrogen, phosphate and silicate (Fig. 6c, d, e). The analysis of these ratios across different seasons (see Appendix A1) reveals that, throughout our time series, winter has moved from a potential phosphate-limiting situation towards a slight nitrogen-limiting system. Additionally, the year 2014 exhibits indications of potential phosphate limitation across all seasons. Furthermore, autumn 2012 and, to a lesser extent, spring 2013 and summer 2018 demonstrate signs of potential phosphate limitation as well. These seasonal variations are well reflected in the annual nutrient ratio (Fig. 7).

### 3.3.2 Phytoplankton inter-annual dynamics

Environmental changes observed during the decadal survey have directly affected the biomass, abundance and composition of phytoplankton communities. The integrated analysis, combining total chlorophyll *a*, as a proxy of phytoplankton biomass and total abundance, revealed distinct patterns (Fig. 8). Chlorophyll *a* showed a succession of increasing and decreasing phases (Fig. 8a). The initial increase in biomass was notably influenced by the peak of 10.70 $\mu$g L$^{-1}$ in February 2013. During the same



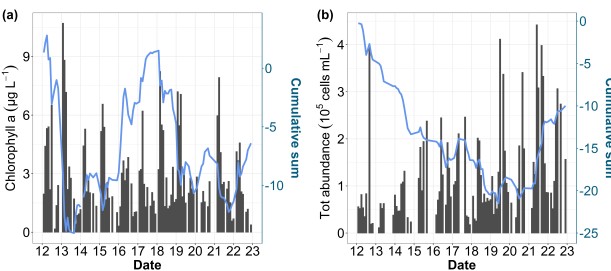

**Figure 8.** Time series of total phytoplankton biomass using (a) chlorophyll *a* and (b) total phytoplankton abundance. Bar plot represents monthly raw data (left y-axis). The blue lines represent the cumulative sum of anomalies over time (right y-axis).

year, phytoplankton abundance was remarkably low (Fig. 8b). After this phase, the chlorophyll *a* time series showed a decline, notably due to a weak spring bloom in 2016 towards higher values in 2018 and 2021. Despite these fluctuations, statistical tests
on chlorophyll *a* concentration revealed no significant decadal trend (Table 4). Conversely, while total cell abundance showed interannual fluctuations with maximum values over $4 \cdot 10^6$ cell mL$^{-1}$ in 2019 and 2021, analysis of raw data, cumulative sums and the Mann-Kendall test indicated a significant increase over the last decade (Fig. 8b and Table 4).

Over the past decade, notable changes in the structure of phytoplankton communities have been observed. Examination of raw data (Fig. 9, black bars) reveals pronounced seasonality, characterized by alternating periods of high and low abundance
across all groups. This seasonality, broken down by group, is further presented in Fig. 5. Cumulative sums of various phytoplankton groups indicated a decadal increase in the abundance of OraPicoProk, RedPico, and OraNano over the period of our study (Fig. 9a, b, c). In 2021, OraPicoProk and RedPico exhibited their highest abundance. Furthermore, the high abundance recorded since 2019 for RedPico, significantly influenced the trends of these groups as well as the total phytoplankton abundance. Statistical trend analysis confirms a significant decadal increase in abundance of the latter two groups for all stations, as
well as for total abundance (Table 4). OraNano depicted a clear trend for some nearshore and offshore stations as well, whereas a non-significant increase characterized frontal and offshore waters. The cumulative sum for RedNano depicts successive phases of increase until 2016, notably attributed to a robust bloom in 2015, followed by a decline until the series' conclusion (Fig. 9d). In spite of a more or less important decadal increase estimated in all stations, no significant trends were evidenced for RedNano. Conversely, RedMicro showed an overall decreasing trend, particularly evident since 2016, a significant trend
was further confirmed at the most coastal stations (Fig. 9e, Table 4).

## 4 Discussion

Between 2012 and 2022, the DYPHYRAD time-series showed a significant decadal increase in temperature, changes in nutrient balance and an increase in phytoplankton abundance due to higher contribution of smaller cells. Spatial and temporal variability along the transect was highlighted throughout the decade, with a more or less pronounced spatial gradient from





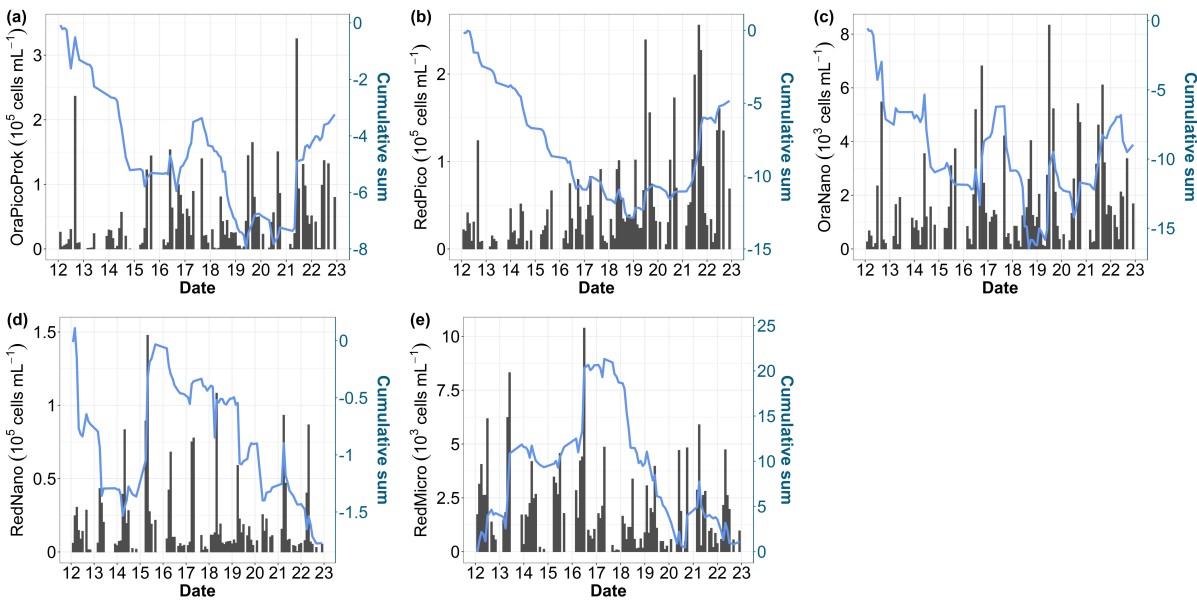

**Figure 9.** Time series of (a) total abundance, (b) OraPicoProk, (c) RedPico, (d) OraNano, (e) RedNano and (f) RedMicro. Bar plot represents monthly raw data (left y-axis). The blue lines represent the cumulative sum of anomalies over time (right y-axis).

**Table 4.** Trends and magnitude of change in phytoplankton chlorophyll *a*, total and functional groups abundance (cell mL$^{-1}$) determined by flow cytometry. Bold indicates that the trend was significant (p-value < 0.05) over the period 2012-2022.

| Phytoplankton biomass/abundance | R0 | R0' | R1 | R1' | R2 | R2' | R3 | R3' | R4 |
|---|---|---|---|---|---|---|---|---|---|
| Chlorophyll *a* ($\mu$g L$^{-1}$) | -0.034 | -0.101 | -0.096 | +0.165 | +0.248 | -0.021 | -0.104 | -0.116 | -0.386 |
| Total abundance (cell mL$^{-1}$) | **+8,214** | **+11,205** | **+8,542** | **+8,499** | **+9,983** | **+12,692** | **+15,446** | **+12,014** | **+9,278** |
| OraPicoProk (cell mL$^{-1}$) | **+1,898** | **+2,550** | **+2,181** | +2,223 | **+3,260** | **+3,921** | **+5,008** | **+4,075** | **+3,543** |
| RedPico (cell mL$^{-1}$) | **+4,195** | **+6,345** | **+5,290** | **+5,372** | **+5,534** | **+7,731** | **+8,270** | **+7,901** | **+6,120** |
| OraNano (cell mL$^{-1}$) | **+58** | +41 | **+73** | +22 | +22 | +51 | **+71** | **+66** | +68 |
| RedNano (cell mL$^{-1}$) | +5 | +221 | + 32 | +81 | +50 | +189 | +311 | +113 | -161 |
| RedMicro (cell mL$^{-1}$) | **-122** | **-130** | -65 | -106 | -43 | -71 | -24 | -50 | -25 |

nearshore to offshore waters. We also identified for the first time a spatial and seasonal pattern of the environmental parameters, phytoplankton biomass and abundance of the six phytoplankton functional groups in the EEC.

## 4.1 Changes in physical parameters

The observed decadal increase in sea surface temperature, considering monthly mean values along the transect, was in line with other studies carried out on a larger temporal and/or spatial scale in the English Channel (Cornes et al., 2023; McLean



et al., 2019; Saulquin and Gohin, 2010). While some longer-term investigations (spanning 30 years) failed to demonstrate this temperature rise (Lefebvre and Devreker, 2023; Tinker et al., 2020), Tinker et al. (2020) highlighted specific years, such as 2014, 2015 and 2017, as amongst the hottest ones on record for sea surface temperature over the past 125 years in the EEC region. Data for the year 2022 were not included in these earlier analyses, yet Simon et al. (2023) highlighted a pronounced marine heatwave in 2022, closely associated with exceptionally high temperatures recorded during that summer (Guinaldo et al., 2023). This phenomenon was also recorded in our time series, which could further corroborate the trend towards increasing SST. This trend is likely to be consolidated in the coming years, as 2023 ranks as the second hottest year since 1991 (Météo-France). It is noteworthy that the influence of the Atlantic Multidecadal Oscillation (AMO), elucidated by Kerr (2000), has been acknowledged for temperature variations in the English Channel (Edwards et al., 2013; Auber et al., 2017). However, despite these natural oscillations, the overall rise in water temperature may directly impact physico-chemical characteristics of water masses along the DYPHYRAD transect and affects its resident phytoplankton organisms (Richardson and Schoeman, 2004). Regarding salinity, our study did not reveal any significant trends, even though the study of cumulative sums revealed a period of increasing salinity extending from winter 2014 to winter 2019, followed by a slight decrease during the last years. Salinity is a relatively stable physico-chemical parameter, however even the slightest change can have significant implications for the marine environment. Station BL1 (50°43'90 N; 1°33'00 E), located around 5.6 km south of our study area, has shown an increase in salinity since 1992 (Lefebvre and Devreker, 2023; Hernández-Fariñas et al., 2014). Increasing sea surface salinity can be attributed to the combined effects of rising sea surface temperatures and significant reduction in river flows between 1998 and 2019, particularly of the Seine and Somme rivers (Huguet et al., 2024).

## 4.2 Change in nutrients concentrations and ratio

In the current study, the long-term trends in inorganic nutrients, essential for phytoplankton growth showed contrasting results. The trend analysis showed an increase in the monthly mean of nitrogen ($[NO_2 + NO_3]$) concentrations in the transect. On the other hand, the spatial trend analysis showed a decrease in nitrogen (at least in nearshore waters) during the studied period. This decline confirmed substantiated by observations along Boulogne-sur-Mer (transect BL), where nitrogen concentration exhibited a consistent decrease ranging from -0.63 to -1.72 $\mu$M over the two-decade period (2000-2020 Lheureux et al., 2023). The dominant forms of dissolved nitrogen in the EEC are nitrite and nitrate, and are strongly influenced by continental inputs as well as Atlantic offshore inputs and atmospheric deposition (Dulière et al., 2019) and showed significant seasonal variability. The greater reduction in dissolved inorganic nitrogen in nearshore compared to offshore waters could therefore be explained by a reduction in continental inputs. The winter 2013-2014 emerged as a remarkable period during which nitrogen concentration was the highest. This is especially meaningful in the context of the extreme weather events of winter 2013-2014, characterized by strong storm events and unprecedented rainfall, resulting in remarkably high turbidity levels (Matthews et al., 2014; Masselink et al., 2016; Gohin et al., 2015). Temporal trends on the cumulative sums revealed a steady decrease in silicate concentration, while spatial trends indicated increasing silicate levels in frontal and offshore waters. A previous study at the SRN station BL1 (off Boulogne sur Mer harbour) showed a significant increase in silicate concentrations between 1992 and 2021 (Lefebvre and Devreker, 2023), which could be explained by the location and different sampling conditions. Silicate dynamics are linked to





the weathering of rocks, 80 % of which are introduced into the ocean by rivers (Conley, 2002) and play a central role as the
main limiting nutrient in the area (Lefebvre et al., 2011). Notably, diatoms, a key phytoplankton group, have shown a strong
positive association with silicates availability and dissolved inorganic nitrogen (DIN; Leynaert et al., 2002; Hernández-Fariñas
et al., 2014). During this study, our trend analysis showed an increase in phosphate (in nearshore waters) and a more complex
pattern of variability in the cumulative sums with alternating increase and decrease phases. The high phosphate concentration
was particularly notable in winter 2015 and 2019. According to Lefebvre et al. (2011), phosphate is the second most limiting
nutrient after silica in the EEC. These changes in nutrient concentration over the decade can led to different potential resources
limitations in the environment and, therefore, affect composition and dynamics of phytoplankton communities. Analysis of
these annual limitations over time (Fig. 7) has shown that the ecosystem is not limited by nitrogen, unlike temperate coastal
region where nitrogen generally limits primary production (Blomqvist et al., 2004). However, this limitation varies greatly
with seasons and has a consequence for the succession of the phytoplankton communities. The imbalance associated with high
nitrogen concentrations (although $NH_4^+$ not measured in the present study) may be due to terrigenous inputs of nitrogen and
phosphorus from intensive agriculture (Garnier et al., 2019). However, the decline in nitrogen levels in nearshore waters over
the past decade can be attributed to the reduction in river flows along the French coast (Huguet et al., 2024) resulting in the
application of European Directives (WFD and MSFD) that aimed at reducing nitrogen and phosphorus inputs into aquatic
systems (Vigiak et al., 2023). Conversely, rising temperatures may lead to increased phosphate release from sediments, which
could explain the rise in phosphate concentrations despite attenuation efforts (Wu et al., 2014; Vigiak et al., 2023). In addition,
Lheureux et al. (2023) documented an increase in Si:P and Si:N ratios at the SRN station in Boulogne-sur-Mer, while the
SOMLIT (National Observation Service of the Research Infrastructure ILICO) coastal station (South of Boulogne-sur-Mer
and further offshore) showed a decrease in all N:P, Si:P and Si:N ratios. In our study, we observed similar trends in nutrient
ratios as observed for the SOMLIT coastal station, with a decrease in ratios over the last decade.

## 360    4.3    Changes in phytoplankton biomass and abundance

Rising temperatures, decreasing annual river flows, nitrogen depletion and nutrient imbalances can lead to a decline in phy-
toplankton biomass, primary production and certain phytoplankton communities such as diatoms as shown in recent studies
in the North Sea and English Channel (Holland et al., 2023b; Breton et al., 2022; Capuzzo et al., 2018). A decline in chloro-
phyll *a* (as a proxy of phytoplankton biomass) has already been described in the EEC using satellite images for the last two
decades (Huguet et al., 2024; Gohin et al., 2019). No significant trend in chlorophyll *a* could be identified in the present work,
however the cumulative sums did reveal successive phases after an initial decrease in chlorophyll *a* until 2013, followed by
an increase until 2018 and then a decrease until 2022. The year 2013, in addition to being the coldest in the series, also pre-
sented a potential limitation in phosphate and silicate, which can be detrimental to the growth of large phytoplankton groups
(e.g. most diatoms). The year 2018 is also a year of silicate and nitrogen limitation. These years both followed years of lower
riverine input to the Somme (2012 and 2017; HydroPortail, https://www.hydro.eaufrance.fr/), which may have influenced nu-
trient inputs in subsequent years. In contrasting ways, high nitrogen concentration (including $NH_4^+$), phosphorus limitation or
environmental nutrient imbalances favor haptophyte species such as *Phaeocystis globosa*, able to take advantage of remaining





resources (Tungaraza et al., 2003; Lancelot et al., 2011; Lefebvre et al., 2011; Breton et al., 2022). These results align with seasonal observations, including the phosphate-limited spring of 2015 and the RedNano bloom peak recorded in our study. In
the Southern North Sea and the EEC, RedNano is known to be largely dominated by *Phaeocystis globosa* during spring blooms (Guiselin, 2010; Thyssen et al., 2015; Bonato et al., 2015; Louchart et al., 2024). The decline in nearshore nitrogen levels and the return to nearly equilibrium values of the N:Si:P (16:16:1) ratio since 2016 could potentially reduce *Phaeocystis globosa* blooms, to the benefit of other species or functional groups, consistent with trends observed in RedNano cumulative sums. The positive correlation between nitrogen availability, particularly during winter, and chlorophyll *a* concentration has been corrob-
orated. Dissolved Inorganic Nitrogen (DIN) decrease during the time-series (1994-2018) was associated with an increase in diatoms (notably *Pseudo-nitzchiza*) and a decrease in *Phaeocystis globosa* in the high DIN concentration area (Lefebvre and Dezécache, 2020). If such a trend in nitrite and nitrate concentrations persists, spring blooms of *Phaeocystis globosa* could be reduced.

The use of flow cytometry for long-term *in vivo* monitoring of phytoplankton communities has revealed a change in size
structure. The EEC experienced a significant increase in the total abundance of cells mainly because of picoeukaryotes and *Synechococcus* spp.. As absolute and relative abundance of picophytoplankton increased, nano– (to some extent) and microphytoplankton decreased. This phenomenon of microphytoplankton decreasing to the benefit of small cells, in particular *Synechococcus* spp. cyanobacteria, was described by Schmidt et al. (2020) in the Western English Channel (L4 station, 2007-2018). These cells are indeed more competitive under low nutrient conditions (Sommer et al., 2017) because of lower resource
requirement and higher Surface:Volume ratio. However, they are less nutritious primary producers of higher food webs organisms, which can lead to a decline in higher trophic levels (Schmidt et al., 2020; Holland et al., 2023b).

## 4.4 Phytoplankton variability dominated by seasonality

Our study showed that seasons explained more than 50 % of the variability observed in phytoplankton communities. This seasonality has been described in numerous studies, notably in regards to the spring bloom of *Phaeocystis globosa*, which can
account for up to 80 % of the total phytoplankton biomass in the EEC (Bonato et al., 2016; Guiselin, 2010). The rest of the time, phytoplankton biomass determined by microscope observation (thus excluding picophytoplankton and small nanophytoplankton) is mainly dominated by diatoms and can reach 85 % of total phytoplankton biomass (Breton et al., 2000; Lefebvre et al., 2011; Hernández-Fariñas et al., 2014). In terms of abundance, cyanobacteria, picoeucaryotes and *Phaeocystis globosa* dominate the area (Bonato et al., 2016). Winter and summer periods are dominated, in terms of abundance, by *Synechococcus* spp.
and picoeucaryotes (Bonato et al., 2016), although they may not share the same niches/habitats (Louchart et al., 2024) which allow them to bloom at the same period (Fig. 5). Other groups are also present in lower abundances, such as cryptophytes, coccolithophores and dinoflagellates (Hernández-Fariñas et al., 2014; Bonato et al., 2016). The spatial gradient is present for most groups and strongly marked for nano- and microphytoplankton, with abundances sometimes three times higher at the coast especially during bloom periods because there are more resources available nearshore (due to the inputs from the rivers).
During the autumn-winter period, changes in spatial community structure was observed with lower abundance of dominant chlorophyll *a* nanophytoplankton at nearshore (R0) and offshore (R4) stations than at frontal stations (R1'and R2). This spa-





tial conformation can be explained by the action of coastal flow on water bodies, as well as by the action of tides and wind speed and direction (Brylinski et al., 1991; Sentchev and Yaremchuk, 2007). This accumulation of nanophytoplankton in the frontal zones has already been shown in the southern North Sea to be linked to the greater presence of nutrients in these struc-

tures than in other bodies of water (Gieskes et al., 2007). Our study has also highlighted the importance of bottom-up control in phytoplankton abundance and biomass distribution, but other parameters such as zooplankton predation (Cotonnec et al., 2001; Breton et al., 2021) and seasonal bacterial/microbial and viral interactions can play a significant role in phytoplankton community variability (Brussaard, 2004; Lamy et al., 2009).

## 4.5   General discussion, limitations and perspectives

The sampling strategy within DYPHYRAD surveys allowed acquiring additional data at higher sampling frequencies and finer spatial scales than other monitoring networks of phytoplankton-related variables (SOMLIT, SRN-REPHY, PHYTOBS). If our approach is also characterized by a fine spatial resolution, its temporal resolution is lower than high frequency moorings or automated stations of French national Coast-HF network (as the MAREL-Carnot automated station off Boulogne-sur-Mer; Halawi Ghosn et al., 2023). Moreover, our surveys made it possible to decouple stations in order to account for the entire

coastal-offshore gradient – a frontal zone separating waters influenced by desalination from river inputs and offshore waters under a macrotidal regime, accounting for tidal variability. Most long-term studies on the evolution of phytoplankton communities over time are either based on the evolution of chlorophyll *a* as a proxy for phytoplankton biomass to explain changes linked to environmental parameters, or based on taxonomical phytoplankton counts by microscopy. However, this kind of approach does not seem sufficient, as it neglects the influence of smaller groups (e.g. picoeukaryotes, cyanobacteria, small nanophyto-

plankton; McQuatters-Gollop et al., 2024) which play an essential role in food webs. The advantage of automated pulse-shape flow cytometry is that the methodology is the same for analysis of the entire phytoplankton size range (Dubelaar et al., 2004), *in vivo*, avoiding any damage or effect of fixatives. The optical characteristics of each particle can then be used to monitor not only abundance, but also functional traits specific to each phytoplankton functional group (Fontana et al., 2018; Fragoso et al., 2019; Louchart et al., 2020). In order to exploit these features and upscale such results over the long term, it remains essential

to improve standard operating procedures for better intercomparability and interoperability between machines, work still in progress in the frame of current international projects as JERICO S3 and OBAMA NEXT.

## 5   Conclusions

This local-scale study showed an increase in sea surface temperature, nearshore phosphate and offshore silicate, as well as a decrease in nitrogen (nitrite and nitrate) concentration in nearshore water over the last decade. Pulse-shape flow cytometry time

series allowed to explore the spatio-temporal *in vivo* dynamics of almost the whole phytoplankton community. A significant increase in small phytoplankton (including cyanobacteria) and a decrease in microphytoplankton abundance (in coastal water) were evidenced. While our time series is too short to draw definitive conclusions about long-term and complex climate change impacts, it allows us to make an initial assessment of change within phytoplankton communities in the EEC by the Strait





of Dover. It is crucial to sustain sampling efforts using automated techniques like flow cytometry to monitor exhaustively
the evolution of phytoplankton dynamics. This monitoring should be integrated as a complement of existing low-frequency
reference national and regional observation networks, and incorporated into high-frequency surveyx as carried out in short-
term previous studies on automated stations (Thyssen et al., 2014; Robache, in prep), ships of opportunity (Marrec et al.,
2021) and oceanographic cruises (Bonato et al., 2016; Louchart et al., 2020, 2024). Supported by a more comprehensive
characterization of PFGs, this approach will greatly enhance our understanding of the impacts of global and anthropogenic
changes on phytoplankton functional diversity. Moreover, when coupled with productivity measurements (Aardema et al.,
2019) and integrated into predictive models, it becomes possible to evaluate the potentialities of food web evlution and overall
ecosystem functioning.

## Appendix A: Seasonal nutrients limitation

To study the evolution of nutrient limitation in more detail within the annual evolution, we analyzed seasonal N:P:Si ratios
(Nitrogen/Phosphate/Silicate; Fig. A1). This representation shows a near-constancy in winters with little or no potential silicate
limitation and slight phosphate or nitrogen limitation. Spring shows a potential phosphate limitation for the years 2013, 2014,
2015 and 2018, while the other years do not seem to be limited or slightly limited by silicate or nitrogen. Autumn 2015 and
2020 appear to be potentially silicate-limiting, whereas 2012 and 2014 show a clear phosphate limitation. Summer 2014 and
2018 tend to be slightly potentially phosphate-limiting, while summer 2013, 2015 and 2016 tend to be potentially silicate-
limiting. The other summers do not appear to be limiting or at least slightly potentially nitrogen limiting, according to the data
presented here.

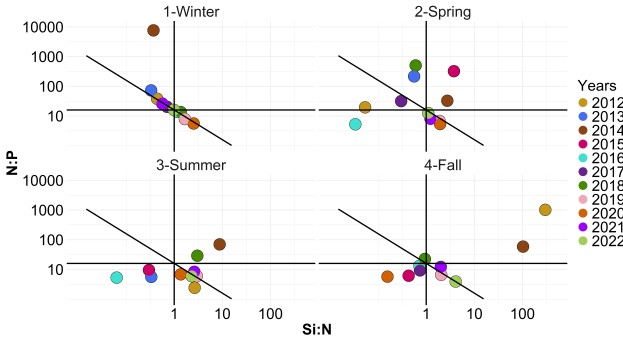

**Figure A1.** Evolution over seasons of potential N:P:Si nutrient limitations according to the Redfield molar ratio (C:N:Si:P = 106:16:16:1;
Redfield et al., 1963). The horizontal line represents the N:P limit of 16:1, the vertical line the Si:N ratio of 1:1 and the diagonal line the
Si:P = 16:1 ratio. The colors correspond to the average ratio for each year.

*Data availability.* A data paper (Hubert, in prep) is in preparation. Data will be made available on request.



*Author contributions.* ZH, LFA and SM conceived and designed the study. ZH performed the data treatment, the code and the analysis of data under the supervision of SM and LFA and advise of AE, KR (figure optimization) and AL (GAM analysis). CG, VC, MC and EL
contributed to data collection and production. ZH wrote the first manuscript draft and all authors contributed to the final version.

*Competing interests.* The authors declare that they have no known competing financial interests or personal relationships that could have appeared to influence the work reported in this paper.

*Acknowledgements.* We would like to thank the crew (Christophe Routtier and Noël Lefilliatre) of the research vessel *Sepia II* (CNRS INSU, French National Oceanographic Fleet) as well as Eric Lécuyer for CTD data and also students and interns of LOG which helped in the
collection and production of environmental data since 2012. We are also grateful to Eric Lécuyer for his constructive remarks to improve the discussion. ZH is supported by a Doctorate grant from the Région Hauts-de-France and Université du Littoral Côte d'Opale (ULCO) and the Doctoral School ED STS (UPJV, UA, ULCO). DYPHYRAD (P.I. LFA) monitoring of phytoplankton was initiated with the support of the INTERREG IV A "2 Seas" DYMAPHY project (2010-2014, P.I. LFA), followed by the JERICO NEXT (2015-2019) and S3 (2020-2024) H2020 European projects. Monitoring was also supported by French State and Nord-Pas-de-Calais–Hauts-de-France Regional Contracts
(CPER) *Phaeocystis* Bloom (2001-2008) and MARCO (2015-2021), which supported the purchase of CytoSense and CytoSub sensors and by CPER IDEAL and its Observation, Geomatic and Remote Sensing technical platform (2021-2028). This work is also supported by the Priority Research Project "Ocean and Climate" PPR RioMAR supported by a France 2030 grant (ANR-22-POE-0006) and the Graduate School IFSEA that also benefits from a France 2030 grant (ANR-21-EXES-0011) operated by the French National Research Agency. Part of this work was supported by the JERICO-S3 project, receiving funding from the European Union's Horizon 2020 research and innovation
programme under grant agreement n° 871153 and by the OBAMA NEXT project under European Union's grant agreement 101081642.



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
