# Peer review of "Decadal changes in phytoplankton functional composition in the Eastern English Channel: upcoming major effects of climate change?"

_EGUsphere, 2024_

## Referee Comment (RC1)

**Review of the manuscript entitled "*Decadal changes in phytoplankton functional composition in the Eastern English Channel: evidence of upcoming major effects of climate change?*" by Hubert et al.**

**General comments**

The manuscript by Hubert et al presents a very interesting dataset on a coastal ecosystem, combining environmental data with data on phytoplankton functional groups. Such a time series on flow cytometry in combination with environmental data at high temporal resolution over multiple years is relatively rare and can give a detailed view into the variability of the phytoplankton community over the last decade. The results of this manuscript are thereby very interesting and fit very well within the scope of the journal. The paper is generally coherent and well written, the references are complete and the study is embedded in a large body of literature on the region. The data analysis is thorough, approaching the dataset from multiple angles, but could be reported more completely and visualized more concisely, which are discussed in more detail below.

**Specific comments**

- Title: The question posed in the title whether there is evidence for major effects due to climate change is a bit vague (did you find evidence or not?). I guess it could be a valid question when 'evidence of' is removed from the sentence, although still speculative and not clearly addressed in the main text. Although in the conclusion is rightfully mentioned that the time series is too short to draw definitive conclusions about long-term and complex climate change, a paragraph discussing which evidence there is to suspect major effects due to climate change would improve the manuscript.
- Methods:
  - o Line 87: cooled how? Do you have a temperature range and a maximum time from sampling until freezing?
  - o Line 92: Using the term nitrogen to refer to [$NO_3^- + NO_2^-$] is a bit confusing. Nitrogen is used to refer to the chemical element itself, while $NO_3^-$ and $NO_2^-$ are just a subset of the nitrogen compounds that are present in seawater. Also, in some figures and text the term NOx is used instead of nitrogen, which is generally used to refer to nitric oxide and nitrogen oxide, not nitrate and nitrite. I would suggest to be concise (and consistent) and refer to the property as [$NO_3^- + NO_2^-$].
  - o Line 100: How many different instruments? Please specify how you addressed comparability between instruments.
  - o Line 121: Please specify brand and manufacturer of the beads.
  - o Section 2.4.1: which k did you use (and why), which type of smooth (cubic spline?). It could be interesting to add the stations as grouping level to test whether station-level variability is significant (see Pedersen et al., 2019 on Hierarchical generalized additive models in ecology: an introduction with mgcv).
  - o Section 2.4.4: nutrient imbalance is a bit suggestive, maybe nutrient stoichiometry?
- Results:
  - o It would be good to include more data on the statistics, for instance by adding result tables to the appendix. This especially concerns; the number of datapoints used in the statistical analysis (separated per station), the GAM results, the Tukey post-hoc test of the PERMANOVA.

- o Throughout the result section it would be good to mention more of the actual measured values in text (for instance in line 195 and 200).
- o Line 195-204: This discrepancy between chlorophyll and cell abundance is quite remarkable, and clearly shows that just using chlorophyll to inform on the phytoplankton community gives an incomplete view of the phytoplankton community. It would be interesting to see to what extend this is due to a variability in pigments per cell as a light acclimation strategy or due to a community shift and cell size, for instance by plotting the amount of fluorescence per cell volume (TFLR/TFWS or TFLO/TFWS).
- o Line 221-224: If I understand correctly, the 45 and 39% refer to the factor 'season', which would then be the most important explaining factor, not year?
- o Since you are using GAM already, why not extend this to the decadal dynamics to plot a smooth through the time series? I see how the cumulative sum is informative, but a smooth on top of the bar plot would be more intuitive visualization of the temporal trend (as an addition, and just a suggestion).
- o N:P ratio: I really like the visualization, which is a clever way to represent nutrients stoichiometry. However, I do miss seeing the original data (or at least error bars) to evaluate the spread around the average point per year, or at least error bars. Appendix 1 shows that outliers have a high effect on the average plot (the high N:P outlier in the winter of 2014 seems to determine its position in the average plot, while in winter nutrients are not in limiting concentrations, while in the seasons where macronutrients might be limiting the N:P ratio is close to 16 again. On the other hand for the year 2016 the high Si:N in fall seems to drive the position of the point in overall plot).
- - Figure and table captions: need more detail so it is possible to understand the figure without having to refer to the main text. Please include a brief description of the methods/statistics that are visualised in the figure. Also include the data that is visualised (whole dataset/per station/number of datapoints).

- - Figures: need better units and axis, and either confidence intervals or the original data. It would also be nice to plot them stacked in columns, since many share a common x-axis, which would make it easier to compare between figures/parameters.
  - o Figure 3 and 4: add y-axis units, add confidence intervals and/or individual data points
  - o Figure 6: the x-axis needs to be more clear. They are labeled date 12 to 23, based on reading the rest of the paper I assume that it refers to the year 2012 to 2023 but this should really be more evident from the figure itself. Also the caption needs more detail (sea surface temperature? Which stations? Cumulative sum of what?). I like that the colour of the secondary axis tries to match the line colour, but the colour is slightly off.
  - o Table 3: I guess these results are from the Mann-Kendell test? Please specify in caption. Do these values have a confidence interval? And can they be added to the plots?

- - Discussion:
  - o What I am missing here a paragraph bringing the physical parameters and phytoplankton dynamics together (possibly with a focus on climate change) and some more focus on the results of this study. In the introduction it is mentioned that

previous studies addressed decadal variation in larger plankton and chlorophyll, but that the picophytoplankton has sofar been neglected, which is the added value of this study. However, in the discussion only one paragraph (line 384-391) discusses the dynamics in this smallest size group. It would be nice to expand upon this and maybe add a wider perspective by comparing the results with other study locations or modeling studies (e.g. are the results expected and can we expect it in other regions?).

**Technical corrections**

- correspondence address: I guess something went wrong in the correspondence email address.
- line 2: run-off not rain-off
- line 11: Not clear where the +1.063 comes from, I can't find it in table 3, please double-check and if correct, discuss it in the main text.
- Line 11-12: please check sentence grammar.
- Line 12: Rollback?
- line 14-15: This sentence is a bit out-of-the-blue – shifting periods of what?
- line 17: It would be nice if the last sentence of the abstract discusses more about the data presented and potential implications, not about future works.
- Line 71-73: check grammar
- Line 78: This is great resolution and sounds like a very dense dataset, could you report the number of datapoints?
- Line 79: R/V before vessel name
- Line 95: chlorophyll a is not a measure of biomass given variability in Chl:C
- Line 95: which sampling depths?
- Line 170: typo in where
- Throughout the manuscript: eukaryotes with a k not c
- Figure 7 and Figure A1: please use a consistent color scheme (2019 is now different).
- Figure 7: I really like the 6 zones in the figure and (what I understand to be) the sequence of limiting nutrients, but the use of the 'greater than' symbol in the figure is a bit confusing (shouldn't it be reversed 'smaller then'? Or just use another symbol).
- Check consistency when referring to seasons (autumn vs. fall).
- In text it is often referred to temperature, it would be good to be more concise and be clear you are referring to sea surface temperatures (or SST) so it can't be mistaken for air temperature.

---

## Referee Comment (RC2)

**Review of " Decadal changes in phytoplankton functional composition in the Eastern English Channel: evidence of upcoming major effects of climate change?”**

by

**Hubert Z., Louchart A., Robache K., Epinoux A., Gallot C., Cornille V., Crouvoisier M., Monchy S. and Artigas L. F.**

**GENERAL COMMENTS:**

This paper deals with phytoplankton functional groups (PFGs) evolution over a pluri-annual monitoring program assessed by automated 'pulse shape-recording' flow cytometry (CytoSenses) in the French waters of the Eastern English Channel along an inshore-offshore transect. Seasonal, decadal and spatial dynamics are explored in relation to environmental variables (mostly temperature and nutrients) from 2012 to 2022. The whole phytoplankton size-range (from 0.1 to 800 μm width) is assessed by those flow cytometers and *in vivo* discrimination of six main PFGs : OraPicoProk, RedPico, RedNano, HsNano, OraNano, and RedMicro as described in Thyssen et al 2022. Over the studied period, a general increase of surface temperatures is accompanied by a significant increase of total abundances with a higher contribution of small cells (picoeukariotes and picoprokariotes) and a significant decrease of microphytoplankton.

This work presents the originality of being the first pluri-annual study of the whole size-range of the phytoplankton community characterized by one unique method. The advantages of these devices are the uniqueness of the protocols, and therefore the consistency of the results, and the speed with which they can be obtained. However, these techniques are particularly valuable when the results are coupled with optical identification of the species present, such as those traditionally used in observation networks.

However, the manuscript needs some additional revision, overall, the discussion section needs to be reworked and re-structured in order to clearly address the main questions of the study and to discuss their own results in relation with previous studies.

Hence, given the ecological and methodological interests of the study, I would not reject the paper but I suggest the authors to make some major revisions in the paper.

**SPECIFIC COMMENTS:**

**1- Introduction**, p.3, l. 53: it seems to me that Holland et al. (2023b) showed an increase in diatoms and dinoflagellates, not only the small ones.

**2- Materials and Methods**, p.4, l. 89: nutrient samples were kept frozen before analysis. Freezing may cause polymerization of part of the dissolved Si and therefore underestimate silicic acid concentrations in the samples. If needed (very low concentrations), the authors might comment on this in the paper.

**3- Materials and Methods**, p.4, l. 100. Is there a homogeneous analysis of data (same Cytosense, same detector, time of analysis, operator, etc...) from 2012 to 2022? When looking at Fig.9 (Results, p. 15), it seems like there are some shifts... are those due to real abundances shifts or to any methodological bias?

**4- Materials and Methods**, p. 6, l. 130: Was linear time series interpolation performed only in ONE station ("this station") for filling gaps in the time series data set, or in all stations? Please modify if necessary.

**5- Materials and Methods.** Si:N:P diagrams have been already used in the literature to identify potential nutrient limitations of phytoplankton. Please refer to some original or previous papers using these representations.

**6-** The environmental parameters considered by the authors in the papers are temperature, salinity and nutrient concentrations. What about light availability (either irradiance or underwater PAR measurements if available)? Solar radiation should be the same everywhere for the whole transect for one given sampling date but it will vary with seasons, and, moreover, with years.

Furthermore, seawater turbidity at each station will modify light availability and therefore constrain phytoplankton community structure. If such data (PAR or turbidity) is not available, the authors should consider this important limiting factor and at least discuss this point.

Specially, in the Results section (p. 10, paragraph 3.3.1), the decadal evolution of solar radiation would be an important point to be considered.

**7-** On the same line of thinking, water column vertical structure (pycnocline, thermocline...) should be considered in the discussion since this will structure the whole phytoplankton community including the surface.

**8- Results**. A general and important comment on how results are interpreted in terms of "trends". The authors use cumulative sums as well as Mann-Kendall tests to highlight trends. If Mann-Kendall tests are not significant, therefore positive or negative trends (from cumulative sums) are not to be considered and therefore not interpreted. I would suggest to use Mann-Kendall tests for long term trends over the whole studied period, and cumulative sums for a more short-term tendency (seasonal trends) and to evidence eventual shifts. The combination of both methods is not clearly explained.

**9-** Table 3, p. 12. What is a significant trend of 0.000 (R3, phosphate)? Furthermore, the Table caption says "The figures indicate the magnitude of the trend". Do the authors mean "the values"?

**10-** Page 12, l. 257-259: "The combined analysis of raw data, cumulative sums and trend tests facilitated the identification of trends and periods of change in physico-chemical variables". Could you explain how those periods (and shifts?) are identified.

**11-** Figure 7, p. 13: Are all data points (sampling dates) plotted on the figure? If this is the case, the authors should not talk about potential limitation during winter periods when nutrients are very high and phytoplankton production and biomass are low. If all dates are plotted, average nutrients are biased with winter values and potential limitations for productive periods are hidden.

**12-** Page 13, l. 268-272 (about figure 7): Indeed, figure in Appendix A1 shows those nutrient ratios for each season, and it is clear here that winter nutrient ratios are different from the other seasons (although marker colors do not allow to see the chronology).

The authors also state that "winter has moved from a potential phosphate-limiting situation towards a slight nitrogen-limiting system": it would be more accurate to say that "winter has moved from a N excess to a P excess period".

They also state that "These seasonal variations are well reflected in the annual nutrient ratio": I do not agree with this statement, since it seems to me that seasonal variations are not reflected in the annual average.

**13- Discussion**, p. 14, l. 298. Please recall here the results, *i.e.* which groups of "smaller cells" have significant increasing trends, and refer to table 4. Pico- but also Nano- phytoplankton are both smaller than micro-phytoplankton.

**14-** Page 15, l. 300-301: I fail to see the relevance of this comment. Is it a conclusion of the « Results » section ? Is it really essential?

**15-** Page 16, l. 321-322: How are explained the decrease of river flowrates and the increase in nutrient concentrations (P near the coast and Si in open sea)? Which rivers are the most relevant to be considered in this ecosystem? What about rainfall?

**16-** Page 16, l. 326: The authors say that "The trend analysis showed an increase in the monthly mean of nitrogen concentrations in the transect": I don't understand what increase is being referred to? Is it on a spatial gradient? Table 3 shows a decreasing trend in N during the period. Other statements are also contradictory with Table 3 trends.

**17-** Page 16, 4.2 Subsection "Change in nutrients concentrations and ratio": I fail to see the final goal of the nutrient discussion. I fully understand a discussion on trends for the overall period (own data presented in the paper) in relation to possible causes of those trends (from previous literature) and more specifically consequences on the observed FPG trends (own data presented in

the paper). The messages might be blurred because global and seasonal trends are all discussed together. I would suggest the authors to clearly identify the main messages to be drawn from their own data from this study and then re-organize the discussion section according to this.

**18**- Page 17, l. 357-359: "…while the SOMLIT (National Observation Service of the Research Infrastructure ILICO) coastal station (South of Boulogne-sur-Mer and further offshore) showed a decrease in all N:P, Si:P and Si:N ratios." Is there any publication showing such decrease of ratios? No references are cited.

In addition, citing SOMLIT as a national observation network in the text is necessary but its belonging to the ILICO infrastructure should be placed in the acknowledgment section.

**19**- Page 18, l. 361-383: This subsection is entitled "Changes in phytoplankton biomass and abundances". The whole subsection, and those first twenty lines in particular, are much more discussing nutrients in relation to species identified in other papers. Echoing my comment n°17, in my opinion, a single subsection bringing together the discussion about phytoplankton and nutrients would be more appropriate, and focusing on the authors own results on FPG observations and trends from the present study.

**20**- Page 18, l. 379. I do not find any "positive correlation between nitrogen availability, particularly during winter, and chlorophyll a concentration" in the paper. Have correlations been calculated? If so, a table with the corresponding results would be necessary.

**21**- Page 19, l. 410. The authors state that their study "has also highlighted the importance of bottom-up control" but I don't see what they are exactly referring to. Bottom-up control is very large including light, nutrients, temperature, hydrodynamics… The only controlling factor, which, moreover, has already been widely discussed in the paper, is nutrient availability.

**22**- Page 19, "General discussion, limitations and perspectives". I agree that it is important to highlight the great additional data that this local monitoring survey brings to phytoplankton dynamics understanding, in addition to regular national monitoring services that are related to phytoplankton. Is weekly sampling a general rule (and therefore I agree with the higher sampling frequency)? However, COAST-HF network does not acquire phytoplankton data and therefore I do not think that this would be a limitation.

In addition to the advantages of this technique, I would suggest to mention that the major limitation is the lack of identification at a more precise taxonomical level and therefore the need of coupling those Cytosense measurements with optical identifications.

**TECHNICAL CORRECTIONS:**

1- At different places in the paper (abstract, fig. 7 and fig. A1, page 13) the authors make reference to Redfield ratios. Please add when absent the Brzezinski reference (only done in page 7).

2- Page 2, l. 24: replace "higher food webs" by "through the food web" or "to higher trophic levels".

3- Page 4, l. 87: please replace $SiO_2$ by $Si(OH)_4$ when talking about silicic acid (or dissolved silicon). $SiO_2$ is silica, which is particulate silicon (biogenic or opal, or lithogenic). Also, although much more widely accepted in the literature, it would be more accurate to use "dissolved Si" or "silicic acid" or just "Si" instead of "silicate" (Fig. 3 and 6, pages 8, 10, 11, 13, 16, 17, 20, Table 3).

4- Page 8, l. 199: prefer the words "coastal" or "nearshore" to "littoral".

5- Figure 7, p. 13.

➜ Please explain what ">" or "<" signs mean ("more limiting than" or "higher concentrations"?)

➜ It would help understanding if colors used for years are chosen from clear to dark, or any other color gradient to be able to follow the chronology. Same for Appendix figure.

➜ The horizontal black line is not visible (N:P = 16:1)

6- Page 16, l. 260 (and elsewhere). Use "molar ratios" or "nutrient ratios" instead of "Redfield ratios".

7- Page 15, 4.1 section title "Changes in physical parameters": The only physical parameter of the study is temperature. Salinity is a chemical parameter. The authors may use "Physical and chemical parameters" and then discuss salinity and nutrients in the same section.

8- Page 16, l. 323, 4.2 subsection title: Please add "s" (plural) to "ratio" ("Change in nutrients concentrations and ratios")

---

## Author Comment (AC1)

**Decadal changes in phytoplankton functional composition in the Eastern English Channel: evidence of upcoming major effects of climate change?**

—

**Reply to Referees' Comments**

Zéline Hubert, Arnaud Louchart, Kévin Robache, Alexandre Epinoux, Clémentine Gallot, Vincent Cornille, Muriel Crouvoisier, Sébastien Monchy, Luis Felipe Artigas

First of all, we would like to thank Referee 1 (R1) and Referee 2 (R2) for their constructive remarks and insightful comments. In order to best respond to this feedback, the response letter combines R1 and R2's comments in two separate sections: section 1 deals with R1's comments, section 2 covers R2's comments. To make these responses easier to read, the letter has been structured as follows: the referees' comments are shown in blue, the authors' responses in black, and the changes made in the manuscript, in green.

**1 Referee 1 comments (RC1)**

**1.1 Specific comments**

**1.1.1 Title**

*1. The question posed in the title whether there is evidence for major effects due to climate change is a bit vague (did you find evidence or not?). I guess it could be a valid question when 'evidence of' is removed from the sentence, although still speculative and not clearly addressed in the main text. Although in the conclusion is rightfully mentioned that the time series is too short to draw definitive conclusions about long-term and complex climate change, a paragraph discussing which evidence there is to suspect major effects due to climate change would improve the manuscript.*
Thank you for your comment. We made the choice of a question in the title because of the conclusion we draw. However, as stipulated in the commentary, the question remains speculative due to a reduced time analysis. Therefore, we decided to remove the wording "evidence of" from the title that now reads "Decadal changes in phytoplankton functional composition in the Eastern English Channel: upcoming major effects of climate change?". Following RC1's suggestion, the conclusion section has been enriched with discussion on this topic, such as consideration about oceanic and coastal changes on a global scale, the influence of temperature on the size structure of phytoplankton communities and future projections for the study area (lines 431-439).

**1.1.2 Methods**

*2. Line 87: cooled how? Do you have a temperature range and a maximum time from sampling until freezing?*
Samples were cooled by placing them directly in an icebox with ice packs to reduce plankton activity

and metabolism, and subsequently freezed at -20 °C as soon as we were back from the field trip, which corresponded to a maximum of 3 hours after sampling. The cool storage method and maximum duration has been added to the materials and methods section (lines 90-92).

*3. Line 92: Using the term nitrogen to refer to [NO3- + NO2-] is a bit confusing. Nitrogen is used to refer to the chemical element itself, while NO3- and NO2- are just a subset of the nitrogen compounds that are present in seawater. Also, in some figures and text the term NOx is used instead of nitrogen, which is generally used to refer to nitric oxide and nitrogen oxide, not nitrate and nitrite. I would suggest to be concise (and consistent) and refer to the property as [NO3- + NO2-].*

The term "nitrogen" has been replaced by the specificity "$[NO_2^- + NO_3^-]$" throughout the manuscript except when citing bibliographical references or discussing nutrient limitations which most of the time refer to dissolved inorganic nitrogen $[NH_4^+ + NO_2^- + NO_3^-]$.

*4. Line 100: How many different instruments? Please specify how you addressed comparability between instruments.*

During this 11-year time series, the following four instruments were used: CS-2007-15; CS-2016-78; CS-2019-93 and CS-2019-94 (Fig. A).

[Figure]

Figure A: Time series of abundance across cytometer serial number used for *in vivo* analysis along the DYPHYRAD transect from 2011 to 2022. The colors are those of the Phytoplankton Functional Groups (PFGs).

To minimize any bias that might exist between devices, the same protocol has been used as much as possible over the past decade. We analyzed seawater filtered to 0.2 µm with a trigger on red fluorescence to establish the detection limit between noise (cellular and mechanical) and phytoplankton, and to avoid the risk of omitting part of the phytoplankton compartment. Data were processed by the same person to keep the same clustering. In addition, only abundances were used, to avoid the risk of biasing the analysis by the inclusion of highly machine-dependent fluorescence. Intercomparisons were carried out on cell abundances of common samples processed by two instruments before or after being changed, to ensure consistency of counting between instruments. In addition, a clear increase in abundance was registered on the machine that carried out the vast majority of the time series (CS-2007-15, Fig. A). On the other hand, low-frequency sampling can also increase the probability of "outlier" measurements, giving the impression of a shift. So that the reader has access to these details, the main information mentioned here has been added to the materials and methods following "Phytoplankton functional composition was obtained from *in vivo* samples using CytoSenses (Cytobuoy b.v., Netherlands)" with the following paragraph : "Over the 11 years of the time series, four cytometers were used. To ensure maximum comparability between the data from each instrument, the data were acquired using as much as possible the same protocols, then processed by the same person to maintain clustering consistency. In addition, only abundances were used, to avoid the risk of biasing the analysis by the inclusion of highly machine-dependent fluorescence. Abundances were compared on common samples when machines were changed." (lines 105-109).

*5. Line 121: Please specify brand and manufacturer of the beads.*
The text has been modified to mention this point: "*To accurately perform phytoplankton functional group discrimination and labeling, we used 1 and 3 µm fluorescent beads (labelled with yellow and multi-fluorescence dyes, respectively; Fluospheres Carboxylate-Modified, Invitrogen, 1.0 µm, yellow-green fluorescent and Sphero brand beads, Spherotech Inc., 3.0-3.4 µm, bright intensity).*" (lines 127-130).

*6. Section 2.4.1: which k did you use (and why), which type of smooth (cubic spline?). It could be interesting to add the stations as grouping level to test whether station-level variability is significant (see Pedersen et al., 2019 on Hierarchical generalized additive models in ecology: an introduction with mgcv).*
The $k$ parameters used in the GAMs is the $k$ defined by default by the GAM function of the "mgcv" R package with no constraints imposed manually. This allows the function to automatically adapt the complexity of the model according to the data and the penalty criteria, thus favoring a compromise between adjustment and generalization. Comparison with several values of $k$ for environmental variables and phytoplankton functional groups shows this fit in comparison with several values of $k$ (example with RedMicro, Fig. B). Curve smoothing corresponds to the smoothing of the GAM function in the "Smoothed conditional means" package, wich relies on cubic regression spline for smooth, adjusted interpolation of trends without over-fitting.

[Figure]

Figure B: Comparison of $k$ values with unconstrained $k$ values. Here, example of RedMicro curves over the 11-year period on the different sampling stations along the DYPHYRAD transect.

This clarification was provided in the manuscript by the following sentence: "These relationships were created using the mgcv GAM function, without any manually imposed constraints. The smoothing of the curves corresponds to the smoothing of the GAM function in the "Smoothed conditional averages" package." (lines 147-150).

*7. Section 2.4.4: nutrient imbalance is a bit suggestive, maybe nutrient stoichiometry?*

The term "nutrient imbalance" have been replaced by "*nutrient stoichiometry*" (line 180).

**1.1.3 Results**

*8. It would be good to include more data on the statistics, for instance by adding result tables to the appendix. This especially concerns; the number of datapoints used in the statistical analysis (separated per station), the GAM results, the Tukey post-hoc test of the PERMANOVA.*

Thank you for your remark. These informations were indeed lacking. In the initial submission we made the choice of not displaying the Tukey HSD test as the tables are very heavy to understand. For transparency on our statistical results, we choose to display the Tukey HSD post-hoc in the appendices in the form of 2 separate heat-maps (season, year and station comparisons) for environmental parameters and phytoplankton abundance. The number N of each dataset used in the analyses was specified.

*9. Throughout the result section it would be good to mention more of the actual measured values in text (for instance in line 195 and 200).*

The results have been enriched with the measured values to simplify the reader's understanding without the need for systematic reading of the figures. Examples can be found in the lines 195, 201 or

209 with a range of GAM values for salinity, dissolved Si or chlorophyll *a*. For example the following sentence: "Silicic acid (Si) showed a sharp depletion from offshore to nearshore waters with lower concentrations around day 110 (from 0.3 to 0.6 µmol L$^{-1}$), with a notable early increase observed at station R0 around day 200 (July, 19$^{th}$) compared to a later increase in the other stations."

*10. Line 195-204: This discrepancy between chlorophyll and cell abundance is quite remarkable, and clearly shows that just using chlorophyll to inform on the phytoplankton community gives an incomplete view of the phytoplankton community. It would be interesting to see to what extend this is due to a variability in pigments per cell as a light acclimation strategy or due to a community shift and cell size, for instance by plotting the amount of fluorescence per cell volume (TFLR/TFWS or TFLO/TFWS).*

Thank you for this remark. We totally agree that the most bigger the cells or colonies, the more they contribute to the *in vivo* total chlorophyll *a* signal (FLR). Analysis of the amount of red fluorescence in relation to the average cell volume over the year 2022 highlights the contribution of each of the PFGs to the chlorophyll *a* estimated by fluorescence measurement (Fig. C).

[Figure]

Figure C: Relation between Mean FWS Total and Mean FLR Total measured in sub-surface samples along the DYPHYRAD transect for the year 2022. The colored ellipses are estimated using the *t*-distribution with a 95 % confidence interval.

Chlorophyll content per cell volume (or FLR/FWS) appears to have decreased over time (Fig. D. However, these data have not been included in the paper, as there is a potential greater bias in fluorescence between the different machines than for abundance.

[Figure]

Figure D: Relation between Mean FLR Total/Mean FWS Total during the past 11 years measured in sub-surface samples along the DYPHYRAD transect for the year 2022.

*11. Line 221-224: If I understand correctly, the 45 and 39% refer to the factor 'season', which would then be the most important explaining factor, not year?*

The reviewer is correct. In fact, intra-annual variability is greater than inter-annual variability. The word "season" has been added to the sentence to make it clearer: Over the last decade, "*seasons had significantly explained 45 % and 39 % of the variances in environmental variables and phytoplankton communities*" (PERMANOVA p-value < 0.05) with a strong difference according to the F-statistic score (lines 234-236).

*12. Since you are using GAM already, why not extend this to the decadal dynamics to plot a smooth through the time series? I see how the cumulative sum is informative, but a smooth on top of the bar plot would be more intuitive visualization of the temporal trend (as an addition, and just a suggestion).*

Thanks for the suggestion. In fact, GAM analysis was also applied to the study of decadal analysis. However, even though GAM analysis does reveal the general trend of the series (see figure below), it does not allow to detect changes in the slope of the series such as break events. Its addition to the cumulative sums enables us to complete the information in a synthetic way (Figure E), but seems perhaps redundant with that obtained by the Mann-Kendall tests. This figure was not added to the manuscript.

[Figure]

Figure E: GAM analysis on environmental time series along the DYPHYRAD transect from 2012 to 2022: (a) SST, (b) salinity, (c) $NO_2^- + NO_3^-$, (d) $H_3PO_4$ and (e) Si. The blue line represent the smoothed data, and the standard deviation is shown by the grey area.

*13. N:P ratio: I really like the visualization, which is a clever way to represent nutrients stoichiometry. However, I do miss seeing the original data (or at least error bars) to evaluate the spread around the average point per year, or at least error bars. Appendix 1 shows that outliers have a high effect on the average plot (the high N:P outlier in the winter of 2014 seems to determine its position in the average plot, while in winter nutrients are not in limiting concentrations, while in the seasons where macronutrients might be limiting the N:P ratio is close to 16 again. On the other hand for the year 2016 the high Si:N in fall seems to drive the position of the point in overall plot).*

Thanks for your comment, in fact, using the average data smoothes out the variability that exists within a year or a season. For this reason, Figure 7 has been replaced by Figure F and Appendix 1 has been replaced by Figure G. This new representation was chosen because the addition of the original data to the averages made the latter unreadable, and the addition of the error bars was also unreadable due to the log scale.

[Figure]

Figure F: Evolution over time of potential N:P:Si nutrient limitations according to the nutrient ratios (C:N:Si:P = 106:16:16:1; Redfield et al., 1963; Brzezinski, 1985). The horizontal black line represents the N:P limit of 16:1, the vertical black line the Si:N ratio of 1:1 and the diagonal black line the Si:P = 16:1 ratio. The red box (a) correspond to the zoom area (b). The (b) large colored dots correspond to the average ratio for each year ($N = 11$), while (a) the small lightened dots correspond to original of each year ($N = 1,015$). Orders A < B mean that B is potentially more limiting than A.

[Figure]

Figure G: Evolution over time of potential N:P:Si nutrient limitations according to the nutrient ratios (C:N:Si:P = 106:16:16:1; Redfield et al., 1963; Brzezinski, 1985). The horizontal black line represents the N:P limit of 16:1, the vertical black line the Si:N ratio of 1:1 and the diagonal black line the Si:P = 16:1 ratio. (a) Small dots correspond to original of each year ($N = 1,015$). (b) The colored dots correspond to the average ratio for each year and season ($N = 44$).

In fact, using averages to synthesize information masks part of the seasonal variability and is strongly influenced by extreme values. This is why Figure 7 and Appendix 1 are complementary and provide the reader with more information on the influence of intra-annual variability on inter-annual variability.

**1.1.4 Figure and table caption**

*14. Need more detail so it is possible to understand the figure without having to refer to the main text. Please include a brief description of the methods/statistics that are visualised in the figure. Also include the data that is visualised (whole dataset/per station/number of datapoints).*

Figure and table captions have been expanded to include the methods and data used, thereby simplifying comprehension and reading. The legend now reads parameter studied, number of observations and method or statistic used.

**1.1.5 Figures**

*15. Need better units and axis, and either confidence intervals or the original data. It would also be nice to plot them stacked in columns, since many share a common x-axis, which would make it easier to compare between figures/parameters.*

Thank you for your comment. Where possible, original data was added to the graphs, notably for nutrient plots (see previous answer). This was not possible for the GAMs, as the excessive number of points made the pattern highlighted by this method unreadable. However, to compensate, a confidence interval was added for each GAM (Figures 3, 4 and 5). We improved plots labels, the banner was transformed into the y-axis title, and units were associated with each title. Sample sizes and methods have also been added to the captions, as requested by the reviewer in previous comment.

*Figure 3 and 4: add y-axis units, add confidence intervals and/or individual data points.*

Thank you for your comment. Figs. 3, 4 and 5 have been redrawn to include confidence intervals. However, the large number of points made it impossible to read the curves correctly, and the version containing only lines was retained. The y-axis now shows units. The units were then removed from the legend to avoid redundancy of information.

*Figure 6: the x-axis needs to be more clear. They are labeled date 12 to 23, based on reading the rest of the paper I assume that it refers to the year 2012 to 2023 but this should really be more evident from the figure itself. Also the caption needs more detail (sea surface temperature? Which stations? Cumulative sum of what?). I like that the colour of the secondary axis tries to match the line colour, but the colour is slightly off.*

The labels were homogenized to match the names of the complete years. The y-axis was modified to match the nomenclature of the rest of the manuscript (i.e. "SST", "$[NO_2^- + NO_3^-]$", "$[Si(OH)_4]$" and "$[H_3PO_4]$"). The legend was completed to specify what the data presented correspond to for raw data (monthly average for all stations combined) and cumulative sums (based on the difference between these monthly averages and the monthly average for the period). The color of the cumulative sums has also been changed to match that of the curve exactly. This has also been done for Figures 8 and 9.

*Table 3: I guess these results are from the Mann-Kendell test? Please specify in caption. Do these values have a confidence interval? And can they be added to the plots?*

We have specified in the caption the statistic test that was used by adding "*from Mann-Kendall test and Sen slope calculation*". We did this for caption of Tables 3 and 4.

**1.1.6 Discussion**

*What I am missing here a paragraph bringing the physical parameters and phytoplankton dynamics together (possibly with a focus on climate change) and some more focus on the results of this study. In the introduction it is mentioned that previous studies addressed decadal variation in larger plankton and chlorophyll, but that the picophytoplankton has sofar been neglected, which is the added value of this study. However, in the discussion only one paragraph (line 384-391) discusses the dynamics in this smallest size group. It would be nice to expand upon this and maybe add a wider perspective by comparing the results with other study locations or modeling studies (e.g. are the results expected and can we expect it in other regions?).*

The discussion was restructured to first describe long-term environmental trends and their potential causes. The second part deals with the consequences and links with phytoplankton communities to better discuss the dynamics of phytoplankton in connection with environmental parameters as well as the dynamics of the smallest groups. For example, we included discussion about temperature, nutrient availability and and phytoplankton composition : "Increasing temperature has a significant effect on the cell size of phytoplankton communities and shift to small species (Zohary et al., 2017; Sommer et al., 2017; Zohary et al., 2021). Combined with a decrease in nutrient availability, this phenomenon is amplified from 4.7% per °C to 46% per °C (Peter and Sommer, 2013)." (line 362-365). In addition, we highlighted the importance of the evolution of picophytoplankton abundance as an indicator of climate change with a potential impact on marine food webs and primary production "Sommer et al. (2017) also predict that with a smaller phytoplankton community, a greater proportion of primary production will benefit the microbial food web, to the detriment of the classic grazing food chain." (lines 380-381). These findings have already been observed in more stratified ecosystems such as the Mediterranean, and are likely to be amplified in the English Channel, given the scenarios modeled for the coming decades (El Hourany et al., 2021; Pörtner et al., 2022).

**1.2 Technical correction**

*– Correspondence address: I guess something went wrong in the correspondence email address.*
Indeed, there was an error in the correspondence address. This is now corrected by the following address: "zeline.hubert@univ-littoral.fr"

*– Line 2: run-off not rain-off*
The term "rain-off" have been replaced by "*run-off*" (line 2).

*– Line 11: Not clear where the +1.063 comes from, I can't find it in table 3, please double-check and if correct, discuss it in the main text.*
Fixed with the values from Table 3. +1.063 have been replaced by "*+1.05*" and +0.929 by "*+0.93*" (line 11).

*– Line 11-12: please check sentence grammar.*
Fixed, the sentence "Changes in nutrient concentrations have led to imbalances in nutrient ratios (N:P:Si) compared to Redfield molar reference ratios though a rollback (2012-2018) to balanced ratios (since 2019)." have been replaced by "*Changes in nutrient concentrations have led to imbalances in nutrient ratios (N:P:Si) relative to reference nutrient ratios. However, a return to balanced ratios has*

*been observed since 2019.*" (lines 11-12).

*– Line 12: Rollback?*

The term "Rollback" have been replaced by "*return*" (line 12).

*– Line 14-15: This sentence is a bit out-of-the-blue – shifting periods of what?*

Fixed, addition of "*Based on analysis of environmental parameters and phytoplankton abundance,*" for the context of shift periods (lines 14-15).

*– Line 17: It would be nice if the last sentence of the abstract discusses more about the data presented and potential implications, not about future works.*

Replacement of the last sentence of the abstract by the following sentence: "*These changes in phytoplankton community, favoring the smallest groups, could lead to a reduction in the productivity of coastal marine ecosystems, that could affect higher trophic levels as well as the entire food web.*" (lines 15-17).

*– Line 71-73: check grammar.*

The sentence "The approach combined high frequency compared to most reference observation networks, and high spatial resolution across all water types, from nearshore to offshore waters, in a frontal system by the Strait of Dover." was replaced by "*The approach combines relatively high frequency with high spatial resolution, complementing most reference observation networks for all types of water, from inshore to offshore, in a frontal system near the Strait of Dover.*" (lines 71-73).

*– Line 78: This is great resolution and sounds like a very dense dataset, could you report the number of datapoints ?*

In order to add information about the dataset, the following sentence has been added: "*The data set consists of 1,835 samples distributed along a longitudinal transect, over 268 sampling dates*" (lines 80-81).

*– Line 79: R/V before vessel name.*

Fixed, vessel name was replace after R/V: "*R/V* Sepia II *(CNRS INSU-FOF)*" (lines 79-80).

*– Line 95: chlorophyll a is not a measure of biomass given variability in Chl:C.*

Thank you for this remark. It is true that chlorophyll $a$ (Chl $a$) does not provide a direct measure of phytoplankton biomass. This value is influenced by environmental factors influencing cell physiology (light, nutrients,...). However, the measurement of Chl $a$ remains a commonly used method for indirectly estimating phytoplankton biomass and for folllowing phytioplankton dynamics (detection of blooms). In our study, we used Chl $a$ as a proxy to monitor long-term changes in the phytoplankton community. While we recognize the limitations of Chl $a$ as a proxy of biomass, it allows us to observe general trends in relation to environmental changes, particularly when other direct indicators of biomass (such as particulate carbon) are not available. Moreover, by integrating complementary data such as flow cytometry (abundance and optical parameters of the whole phytoplankton size range), we reduce the uncertainties associated with variations in the Chl:C ratio and obtain a more complete information about phytoplankton dynamics. Therefore, the sentence "Phytoplankton biomass was estimated through chlorophyll *a* concentration analysis in sub-surface" was replaced by "*Phytoplankton biomass was approached using chlorophyll a concentration analysis in sub-surface waters, even though we aknowledge that there is a variability in Chl:C.*" (lines 98-99)

> *– Line 95: which sampling depths?*

All samples are taken from the sub-surface (1 to 2 meters depth). This information was added to the text (line 87).

> *– Line 170: typo in where*

Fixed, "wheere" has been corrected by "*where*" (line 181).

> *– Throughout the manuscript: eukaryotes with a k not c*

Fixed, all "eukaryotes" terms have been corrected.

> *– Figure 7 and Figure A1: please use a consistent color scheme (2019 is now different).*

Colors of the figure A1 have been homogenized with those shown in Figure 7.

> *– Figure 7: I really like the 6 zones in the figure and (what I understand to be) the sequence of limiting nutrients, but the use of the 'greater than' symbol in the figure is a bit confusing (shouldn't it be reversed 'smaller then'? Or just use another symbol).*

The caption of the figure has been completed to clarify the relationship between nutrients with the following sentence: "*The expression "A < B" mean that B is potentially more limiting than A.*".

> *– Check consistency when referring to seasons (autumn vs. fall).*

Fixed. All "fall" terms have been replaced by "*autumn*".

> *– In text it is often referred to temperature, it would be good to be more concise and be clear you are referring to sea surface temperatures (or SST) so it can't be mistaken for air temperature.*

Fixed, all temperature terms have been replaced by "*Sea Surface Temperature*" or "*SST*".

**2 Referee 2 comments (RC2)**

**2.1 Specific comments**

**2.1.1 Introduction**

> *1. p.3, l. 53: it seems to me that Holland et al. (2023b) showed an increase in diatoms and dinoflagellates, not only the small ones.*

Thanks for your remark. Indeed, it was a mistake, which has now been corrected and the word "small" has been removed from the text (line 53).

**2.1.2 Materials and Methods**

> *2. p.4, l. 89: nutrient samples were kept frozen before analysis. Freezing may cause polymerization of part of the dissolved Si and therefore underestimate silicic acid concentrations in the samples. If needed (very low concentrations), the authors might comment on this in the paper.*

Our analyses are carried out on the same machine as SOMLIT samples and under the same conditions (analysis protocol and quality controls). Intercomparisons were carried out within the SOMLIT network (https://www.somlit.fr/) south of Boulogne-sur-Mer (coastal and offshore stations, with nutrient concentrations analyzed by the same person and the same instrument than DYPHYRAD sampling) between storage at 4°C and -20°C, on identical samples at several given dates. Dissolved Si concentrations were compared using linear regression to characterize the potential difference between measured concentrations (Figure H). The measured slope was close to 1 ($\alpha = 1.02 \pm 0.04$) with a strong $R^2 = 0.93$ and a significant p-value inferior to 0.01.

[Figure]

Figure H: Comparison between Si concentrations measured after conservation at 4 °C and -20 °C, represented by the dots in coastal and offshore waters of the SOMLIT observation network (Boulogne-sur-Mer). The red dashed line represents $y = x$ and the black line represents the affine regression ($y = \alpha x$) estimated for the shown data. The error for $\alpha$ slope value, the associated p-value and the coefficient of determination $R^2$ are also provided.

This nutrient preservation method allows samples to be stored for the duration of the analysis without contamination (Dore et al., 1996). Aminot and Kérouel (2007) recommends this type of storage when the sample cannot be analyzed the same day, which is the case here. They also state that freezing can cause polymerization, especially if the water is not very salty, which is not really the case here, except in the case of strong fluvial inputs from upstream estuaries. However, this specification has been added to the material and method with the associated citation with the following sentence : *This nutrient preservation process is recommended when samples cannot be analyzed on the same day (Aminot and Kérouel, 2007).* (linea 92-93).

*3. p.4, l. 100. Is there a homogeneous analysis of data (same Cytosense,same detector, time of analysis, operator, etc...) from 2012 to 2022? When looking at Fig.9 (Results, p. 15), it seems like there are some shifts... are those due to real abundances shifts or to any methodological bias?*

Thanks for your remark. We were aware of potential biases when using differents instruments. However, we minimised the bias using as much as possible the same protocoles. In addition, we analysed filtered sea water on 0.2 µm with a trigger on the red fluorescence to establish the limit of detection between noise (cellular and mechanical) and phytoplankton. Also, all samples were processed by the same operator to keep consistency in the clustering. The time of analysis was consistently kept at

5 minutes for the picophytoplankton protocole and 10 to 13 minutes for the microphytoplankton protocole. Finally, tests on common samples were carried out before each change of equipment to ensure continuity of counting during measurements. We addressed this response to reviewer 1, comment number 4 and proposed a figure summarizing the use of the different devices during the whole decade (Figure A). Supplementary information has been added in the text to specify this (lines 105-109).

*4. p. 6, l. 130: Was linear time series interpolation performed only in ONE station ("this station") for filling gaps in the time series data set, or in all stations? Please modify if necessary.*
This has been done for each station, the material and method has been corrected by stipulating "*at each station*" instead of "this station" (line 139).

*5. Si:N:P diagrams have been already used in the literature to identify potential nutrient limitations of phytoplankton. Please refer to some original or previous papers using these representations.*
Similar paper references, such as Pannard et al. (2008); Schapira et al. (2008); Akanmu (2018) were cited as support for the graphic representation (lines 184-185).

*6. The environmental parameters considered by the authors in the papers are temperature, salinity and nutrient concentrations. What about light availability (either irradiance or underwater PAR measurements if available)? Solar radiation should be the same everywhere for the whole transect for one given sampling date but it will vary with seasons, and, moreover, with years. Furthermore, seawater turbidity at each station will modify light availability and therefore constrain phytoplankton community structure. If such data (PAR or turbidity) is not available, the authors should consider this important limiting factor and at least discuss this point. Specially, in the Results section (p. 10, paragraph 3.3.1), the decadal evolution of solar radiation would be an important point to be considered.*
Light availability has not been taken into account in this study for several reasons. The first was that PAR is highly dependent on the time of the day, daily cloudy or clearly conditions, and discrete samples even though taken at nearly the same hour of the day, do not correspond to a comparable quantity and quality of light and therefore will not necessarily reflect a ten-year trend. Secondly, while measured for most of the decadal samples, PAR during year 2020 was not recorded due to sensor maintenance. Finally, a comparison of PAR data with those from MAREL-Carnot (geographically close to our sampling sites) in order to correctly define a trend over the last decade could have been considered. However, this was not possible due to a partial absence of data from the station from 2012, followed by a total absence from mid-2013.

*7. On the same line of thinking, water column vertical structure (pycnocline, thermocline...) should be considered in the discussion since this will structure the whole phytoplankton community including the surface.*
The study area is almost vertically unstratified and the water masses homogeneous, in line with previous studies carried out in the area (Pingree and Griffiths, 1978; Tinker et al., 2024; Lefebvre and Devreker, 2023). The Eastern English Channel is an strongly mixed sea, due to strong tidal currents and a large residual current especially in the Strait of Dover.

**2.1.3 Results**

*8. A general and important comment on how results are interpreted in terms of "trends". The authors use cumulative sums as well as Mann-Kendall tests to highlight trends. If Mann-Kendall tests are not significant, therefore positive or negative trends (from cumulative sums) are not to be considered and therefore not interpreted. I would suggest to use Mann-Kendall tests for long term trends over the whole studied period, and cumulative sums for a more short-term tendency (seasonal trends) and to evidence eventual shifts. The combination of both methods is not clearly explained.*

Thank you for your comment and suggestion. We understand the importance of distinguishing between significant long-term trends and short-term seasonal variations. Indeed, the initial aim of our approach was to combine the Mann-Kendall test to detect significant trends over the whole period studied, and to use cumulative sums to highlight possible changes in the data. The method was clarified at the end of the first paragraph "Environmental decadal evolution" (lines 270-273) in reference to comment 10 as well with the following sentence : "The combined analysis of raw data and cumulative sums has enabled us to identify periods of change in physico-chemical variables (according to Regier et al. (2019)), such as the transition between 2013 and 2014, as well as the period from 2018 to 2020, by observing changes in slope. On the other hand, trend tests (Mann-Kendall) and slope calculations (Sen slope estimate) facilitated trend identification and quantification.". Within the manuscript, the inaccuracies presented here have aimed to be reduced as much as possible so that the methodology and the results obtained are clear to the reader: the Mann-Kendall test was exclusively applied for the assessment of significant long-term trends, and avoiding interpreting the results of cumulative sums in terms of positive or negative trends if the Mann-Kendall results are not significant; we used cumulative sums as a tool to identify potential changes or tipping points in the data, without interpreting them as overall trends.

*9. Table 3, p. 12. What is a significant trend of 0.000 (R3, phosphate)? Furthermore, the Table caption says "The figures indicate the magnitude of the trend". Do the authors mean "the values"?*

The significance of the trend is based on the Mann-Kendall test, which calculates the sign of the difference between two values. It only highlights whether or not a trend is present, without quantifying the slope, which, on the other hand, is calculated using Sen's slope estimate, based on the average of all slopes between two points. Consequently, changes within the time-series can lead to a negative trend but a zero slope, particularly in the event of a very weak trend or non-linearity of the series. The world "*slope*" has been added after magnitude to avoid ambiguity for the reader.

*10. Page 12, l. 257-259: "The combined analysis of raw data, cumulative sums and trend tests facilitated the identification of trends and periods of change in physico-chemical variables". Could you explain how those periods (and shifts?) are identified.*

Shift periods were identified according to the method of slope changes in cumulative sums (Regier et al., 2019). The analyses that led to each of the conclusions have been added to the sentence: "*The combined analysis of raw data and cumulative sums has enabled us to identifying periods of change in physico-chemical variables (according to Regier et al., 2019), such as the transition between 2013 and 2014, as well as the period from 2018 to 2020, by observing changes in slope. On the other hand, trend tests (Mann-Kendall) and slope calculations (Sen slope estimate) facilitated trend identification and quantification.*" (lines 270-273).

*11. Figure 7, p. 13: Are all data points (sampling dates) plotted on the figure? If this is the case, the authors should not talk about potential limitation during winter periods when nutrients are very high and phytoplankton production and biomass are low. If all dates are plotted, average nutrients are biased with winter values and potential limitations for productive periods are hidden.*

Figure 7 shows the mean value for all sampling dates in the year. It is complemented by Appendix 1, which presents the average of these points by year and season, to discuss the influence of seasons such as winter on annual balances. Figure G shows that spring and summer display a consistent nutrient limitation within the years. Indeed, as suggested by the reviewer 1, the latter has also been enriched with all the original data, to present the variability within these seasons too (1.1.3. Results, comment 13).

*12. Page 13, l. 268-272 (about figure 7): Indeed, figure in Appendix A1 shows those nutrient ratios for each season, and it is clear here that winter nutrient ratios are different from the other seasons (although marker colors do not allow to see the chronology). The authors also state that "winter has moved from a potential phosphate-limiting situation towards a slight nitrogen-limiting system": it would be more accurate to say that "winter has moved from a N excess to a P excess period".*

Thanks for this remark. Figure I shows the evolution of N and P in winter over the last decade. There has been a slight increase in phosphates (although still potentially limiting), while nitrates have fallen considerably (going from an excess to a slight limitation). The system therefore does not appear to switch to excess phosphate concentration.

[Figure]

Figure I: Evolution of "$[NO_2^- + NO_3^-]$" and "$[H_3PO_4]$" concentrations measured during the winter times series along the DYPHYRAD transect from 2012 to 2022. The dots correspond to the average data for each year and station. The curve corresponds to default Loess smoothing, while the dotted black line represents linear regression.

*They also state that "These seasonal variations are well reflected in the annual nutrient ratio": I do not agree with this statement, since it seems to me that seasonal variations are not reflected in the annual average.*

Regarding the question of seasonal variations and their reflection in annual averages, we agree that

these seasonal variations are not fully captured by annual averages, and we have rephrase this section to avoid confusion. Annual averages effectively smooth out season-specific variations, and we have clarify that our intention is to describe a general trend in nutrient evolution over time, without directly inferring these seasonal variations from the averages. For this reason, the sentence "These seasonal variations are well reflected in the annual nutrient ratio" was removed from the manuscript.

**2.1.4 Discussion**

*13. p. 14, l. 298. Please recall here the results, i.e. which groups of "smaller cells" have significant increasing trends, and refer to table 4. Pico- but also Nano- phytoplankton are both smaller than micro-phytoplankton.*

In view of the other changes, this sentence is no longer part of the manuscript.

*14. Page 15, l. 300-301: I fail to see the relevance of this comment. Is it a conclusion of the « Results » section ? Is it really essential?*

In fact, this sentence was intended to reiterate the main points of the results section, but it doesn't belong here. It was removed.

*15. Page 16, l. 321-322: How are explained the decrease of river flowrates and the increase in nutrient concentrations (P near the coast and Si in open sea)? Which rivers are the most relevant to be considered in this ecosystem? What about rainfall?*

The decrease in river flow over time may be due to a general decrease in rainfall distribution (Figure Jb). However, over the last decade, maximum values have tended to increase over time(Figure Ja and b), which could lead to greater runoff and therefore an increase in concentration of phosphate and dissolved silicate due to intense rainy events.

[Figure]

Figure J: Monthly rainfall at Boulogne-sur-Mer (Météo France data): (a) raw data and (b) boxplot. The red dashed line represents linear regression estimated for the shown data.

In the context of our study of the Strait of Dover, the most relevant rivers to consider are, in order of significance, the Somme, followed by the Canche, Authie, Liane, Wimereux and Slack, which have progressively lower flows. This previous sentence was added to the manuscript (lines 328-330).

*16. Page 16, l. 326: The authors say that "The trend analysis showed an increase in the monthly mean of nitrogen concentrations in the transect": I don't understand what increase is being referred to? Is it on a spatial gradient? Table 3 shows a decreasing trend in N during the period. Other statements are also contradictory with Table 3 trends.*

These statements were based on the observation of cumulative sums, but in fact, as advised by the reviewer, the manuscript is revised so that long-term trends are based solely on the Mann-Kendall test and Sen's slope estimates, while transition periods in these series are highlighted using cumulative sums.

*17. Page 16, 4.2 Subsection "Change in nutrients concentrations and ratio": I fail to see the final goal of the nutrient discussion. I fully understand a discussion on trends for the overall period (own data presented in the paper) in relation to possible causes of those trends (from previous literature) and more specifically consequences on the observed FPG trends (own data presented in the paper). The messages might be blurred because global and seasonal trends are all discussed together. I would suggest the authors to clearly identify the main messages to be drawn from their own data from this study and then re-organize the discussion section according to this.*

Thank you for your comment. Indeed, according to this comment and others made by RC1 and yourself, the discussion was revised in both construction and content to better put the results of this study into perspective. The section on nutrient trends has been added to the other long-term environmental trends in the section now entitled: "Decadal trends in physical and chemical parameters".

*18. Page 17, l. 357-359: "...while the SOMLIT (National Observation Service of the Research Infrastructure ILICO) coastal station (South of Boulogne-sur-Mer and further offshore) showed a decrease in all N:P, Si:P and Si:N ratios." Is there any publication showing such decrease of ratios? No references are cited. In addition, citing SOMLIT as a national observation network in the text is necessary but its belonging to the ILICO infrastructure should be placed in the acknowledgment section.*

Thanks for the remark and suggestion. At the beginning of the sentence quoted above (line 237), Lheureux et al. (2023) is cited as a reference showing this decrease. We have acknowledged both the SOMLIT national French observation network and ILICO Research Infrastructure.

*19. Page 18, l. 361-383: This subsection is entitled "Changes in phytoplankton biomass and abundances". The whole subsection, and those first twenty lines in particular, are much more discussing nutrients in relation to species identified in other papers. Echoing my comment n°17, in my opinion, a single subsection bringing together the discussion about phytoplankton and nutrients would be more appropriate, and focusing on the authors own results on FPG observations and trends from the present study.*

Thank you for this proposal to restructure the discussion. In fact, this comment and comment 17 have been taken into account in the restructuring of the discussion, grouping nutrients and the other environmental long-trend in a single section entitled : "4.1. Decadal trends in physical and chemical parameters". The section discussing the influence on phytoplankton communities has been grouped under a second sub-section entitled "4.2. Consequences on phytoplankton funional groups".

*20. Page 18, l. 379. I do not find any "positive correlation between nitrogen availability, particularly during winter, and chlorophyll a concentration" in the paper. Have correlations been calculated?*

*If so, a table with the corresponding results would be necessary.*

Thanks for this remark. In fact, this statement comes from the study of Lefebvre and Dezecache (2020) quoted in the following sentence referring to the SRN REPHY time series off Boulogne sur Mer (close to our site ; line 390). The reference has been added to this sentence as well. No correlations were calculated for our data.

*21. Page 19, l. 410. The authors state that their study "has also highlighted the importance of bottom-up control" but I don't see what they are exactly referring to. Bottom-up control is very large including light, nutrients, temperature, hydrodynamics... The only controlling factor, which, moreover, has already been widely discussed in the paper, is nutrient availability.*

This formulation was removed to clarify the effect of long-term nutrient changes on PFGs.

*22. Page 19, "General discussion, limitations and perspectives". I agree that it is important to highlight the great additional data that this local monitoring survey brings to phytoplankton dynamics understanding, in addition to regular national monitoring services that are related to phytoplankton. Is weekly sampling a general rule (and therefore I agree with the higher sampling frequency)? However, COAST-HF network does not acquire phytoplankton data and therefore I do not think that this would be a limitation. In addition to the advantages of this technique, I would suggest to mention that the major limitation is the lack of identification at a more precise taxonomical level and therefore the need of coupling those Cytosense measurements with optical identifications.*

Thank you for the remark. Indeed, weekly sampling is a general rule for the DYPHYRAD observatory (systematically planned outings), but it is obviously subject to weather conditions. In any case, this constraint also applies to regular national monitoring services that go out every two weeks on paper, with the potential for weather-related cancellations. On the other hand, chlorophyll *a in vivo* bulk fluorescence is a labelled parameter of the COAST-HF network. It therefore includes a variable linked to phytoplankton monitored at high-frequency, but only in a bulk sense. However, we can clearly see here that, despite the absence of a chlorophyll *a* trend, cell abundances vary significantly across space and time within each PFGs and in total. Moreover, the CytoSense has the ability to acquire photos at the same time as optical measurements, which makes it a very useful tool when working in defining sub-groups within the microphytoplankton size class, based on the combination of images and pulse-shapes, although the methodology requires much more time and expertise and is not yet subject to automation as all automated imaging systems do. Moreover, other limitations remain, first of all on the total amount of images taken with the older machines and the amount of images on focus for all machines. Nevertheless, we agree on the interest in coupling the measurements with automated imaging systems, as microscopic references can be found at stations near our area within the PhytOBS network (on both SOMLIT and SRN-REPHY stations). The following sentence was added to the "General discussion, limitations and perspectives" section: "Although this method enables us to study all phytoplankton, it is not a taxonomic technique (except in the case of microphytoplankton via CytoSense photo acquisition) and could be combined with approaches enabling finer identification." (lines 419-421).

**2.2 Technical corrections**

*– At different places in the paper (abstract, fig. 7 and fig. A1, page 13) the authors make reference to Redfield ratios. Please add when absent the Brzezinski reference (only done in page 7).*

The reference was added to all those ratios where Redfield had been cited.

*– Page 2, l. 24: replace "higher food webs" by "through the food web" or "to higher trophic levels".*
The expression "in higher food webs" have been replaced by "*to higher trophic levels*" (line 24).

*– Page 4, l. 87: please replace SiO2 by Si(OH)4 when talking about silicic acid (or dissolved silicon). SiO2 is silica, which is particulate silicon (biogenic or opal, or lithogenic). Also, although much more widely accepted in the literature, it would be more accurate to use "dissolved Si" or "silicic acid" or just "Si" instead of "silicate" (Fig. 3 and 6, pages 8, 10, 11, 13, 16, 17, 20, Table 3).*
SiO2 have been replaced by "Si(OH)$_4$" in the Materials and Mehtods part. In Figure 3, Figure 6, page 10, 11,13, 16, 17, 20 and Table 3 the word "silicate" has been replaced by "Si".

*– Page 8, l. 199: prefer the words "coastal" or "nearshore" to "littoral".*
Fixed, all "littoral" term have been replaced by "*coastal*" (line 212).

*– Figure 7, p. 13. → Please explain what ">" or "<" signs mean ("more limiting than" or "higher concentrations"?) → It would help understanding if colors used for years are chosen from clear to dark, or any other color gradient to be able to follow the chronology. Same for Appendix figure. → The horizontal black line is not visible (N:P = 16:1)*
The meaning of the signs has been added to the figure caption as follows: "*The expression "A < B" mean that B is potentially more limiting than A.*". There is no specific color gradient, just contrasting colors to distinguish them from one another. The figures were completely redone, all elements are normally visible/readable now.

*– Page 16, l. 260 (and elsewhere). Use "molar ratios" or "nutrient ratios" instead of "Redfield ratios".*
Fixed, all "Redfield ratios" term have been replaced by "*nutrient ratios*".

*– Page 15, 4.1 section title "Changes in physical parameters": The only physical parameter of the study is temperature. Salinity is a chemical parameter. The authors may use "Physical and chemical parameters" and then discuss salinity and nutrients in the same section.*
When the discussion was revised, the title of this section was changed as suggested. The new section title is "Decadal trends in physical and chemical parameters".

*– Page 16, l. 323, 4.2 subsection title: Please add "s" (plural) to "ratio" ("Change in nutrients concentrations and ratios").*
Fixed.

**3    Other changes**

Additional minor modifications and technical corrections have been added to the manuscript and are described below.

**3.1 Minor changes**

**3.2 Technical changes**

– Line 82: the number 9 was spelled out "nine".

– Line 104: CytoSense was singular because it was a brand model.

– Figure 7 was enlarged for easier reading.

**References**

Akanmu, R. (2018), 'Nutrients Dynamics and Trophic Status in A Tropical Ocean off The Lagos Coast, Nigeria', pp. 87–99.

Aminot, A. and Kérouel, R. (2007), *Dosage automatique des nutriments dans les eaux marines: méthodes en flux continu*, Editions Quae.

Dore, J. E., Houlihan, T., Hebel, D. V., Tien, G., Tupas, L. and Karl, D. M. (1996), 'Freezing as a method of sample preservation for the analysis of dissolved inorganic nutrients in seawater', *Marine Chemistry* **53**(3), 173–185.

El Hourany, R., Mejia, C., Faour, G., Crépon, M. and Thiria, S. (2021), 'Evidencing the Impact of Climate Change on the Phytoplankton Community of the Mediterranean Sea Through a Bioregionalization Approach', *Journal of Geophysical Research: Oceans* **126**(4), e2020JC016808.

Lefebvre, A. and Devreker, D. (2023), 'How to learn more about hydrological conditions and phytoplankton dynamics and diversity in the eastern English Channel and the Southern Bight of the North Sea: The Suivi Régional des Nutriments data set (1992–2021)', *Earth System Science Data* **15**(3), 1077–1092.

Lheureux, A., David, V., Del Amo, Y., Soudant, D., Auby, I., Bozec, Y., Conan, P., Ganthy, F., Grégori, G., Lefebvre, A., Leynart, A., Rimmelin-Maury, P., Souchu, P., Vantrepote, V., Blondel, C., Cariou, T., Crispi, O., Cordier, M.-A., Crouvoisier, M., Duquesne, V., Ferreira, S., Garcia, N., Gouriou, L., Grosteffan, E., Le Merrer, Y., Meteigner, C., Retho, M., Tournaire, M.-P. and Savoye, N. (2023), 'Trajectories of nutrients concentrations and ratios in the French coastal ecosystems: 20 years of changes in relation with large-scale and local drivers', *Science of The Total Environment* **857**, 159619.

Pannard, A., Claquin, P., Klein, C., Roy, B. and Véron, B. (2008), 'Short-term variability of the phytoplankton community in coastal ecosystem in response to physical and chemical conditions' changes', *Estuarine Coastal and Shelf Science* **80**, 212–224.

Peter, K. H. and Sommer, U. (2013), 'Phytoplankton Cell Size Reduction in Response to Warming Mediated by Nutrient Limitation', *PLOS ONE* **8**(9), e71528.

Pingree, R. D. and Griffiths, D. K. (1978), 'Tidal fronts on the shelf seas around the British Isles', *Journal of Geophysical Research: Oceans* **83**(C9), 4615–4622.

Pörtner, H.-O., Roberts, D., Tignor, M., Poloczanska, E., Mintenbeck, K., Alegria, A., Craig, M., Langsdorf, S., Löschke, S., Möller, V., Okem, A. and Rama, B., eds (2022), *Climate Change 2022: Impacts, Adaptation and Vulnerability. Contribution of Working Group II to the Sixth Assessment*

*Report of the Intergovernmental Panel on Climate Change*, cambridge university press edn, Cambridge University Press, Cambridge University Press, Cambridge, UK and New York, NY, USA.

Regier, P., Briceño, H. and Boyer, J. N. (2019), 'Analyzing and comparing complex environmental time series using a cumulative sums approach', *MethodsX* **6**, 779–787.

Schapira, M., Vincent, D., Gentilhomme, V. and Seuront, L. (2008), 'Temporal patterns of phytoplankton assemblages, size spectra and diversity during the wane of a Phaeocystis globosa spring bloom in hydrologically contrasted coastal waters', *Journal of the Marine Biological Association of the United Kingdom* **88**(4), 649–662.

Sommer, U., Peter, K. H., Genitsaris, S. and Moustaka-Gouni, M. (2017), 'Do marine phytoplankton follow Bergmann's rule sensu lato?', *Biological Reviews* **92**(2), 1011–1026.

Tinker, J., Palmer, M. D., Harrison, B. J., O'Dea, E., Sexton, D. M. H., Yamazaki, K. and Rostron, J. W. (2024), 'Twenty-first century marine climate projections for the NW European shelf seas based on a perturbed parameter ensemble', *Ocean Science* **20**(3), 835–885.

Zohary, T., Fishbein, T., Shlichter, M. and Naselli-Flores, L. (2017), 'Larger cell or colony size in winter, smaller in summer – a pattern shared by many species of Lake Kinneret phytoplankton', *Inland Waters* **7**(2), 200–209.

Zohary, T., Flaim, G. and Sommer, U. (2021), 'Temperature and the size of freshwater phytoplankton', *Hydrobiologia* **848**(1), 143–155.

---

## Referee Report (RR1)

**Review of " Decadal changes in phytoplankton functional composition in the Eastern English Channel: evidence of upcoming major effects of climate change?"**

by

**Hubert Z., Louchart A., Robache K., Epinoux A., Gallot C., Cornille V., Crouvoisier M., Monchy S. and Artigas L. F.**

Thank you very much to the authors for those important modifications on the paper as well as for their answers. The article reads really better now and in my opinion is an excellent paper.

I would like however to address very minor comments that do not need any further revision.

**1-** Please try to be consistent all over the article on the terms used for silicic acid and other nutrients ; there are still some inconsistencies :

➔ L. 342  "dissolved nitrogen" vs. L. 344 "Dissolved inorganic nitrogen"

➔ Appendix A : "dissolved silicate" / "Si limitation" vs "phosphate and nitrogen limitation" / "Slightly limited by Si or nitrogen" / "Si-limiting"

If you decide to use Si, then use N and P (within the whole paper and not only the appendix). Otherwise use "silicon" (which is the element) as "nitrogen" or "phosphorus"

As a reminder, it would be more accurate to use "dissolved Si" (and dissolved N or P…) or "dissolved silicon" (and dissolved nitrogen or phosphorus) or "silicic acid" or just "Si" (and N or P) instead of "silicate"

**2-** I understand that no PAR no light data could be used for the study and this is not a problem. However, since light is an essential factor, I would recommend the authors to consider this important potential limiting factor by a short comment in the Results section (p. 10, paragraph 3.3.1) and/or in the discussion.

**3-** Some sentences remained truncated after the revision:

L. 340 "This decline confirmed substantiated by observations along Boulogne-sur-Mer (transect BL), where $[NO^{-}2 + NO^{-}3]$ concentration exhibited a consistent decrease ranging from -0.63 to -1.72 µM over the two decade period (2000-2020 Lheureux et al., 2023)."

L 370 "Indeed, Lefebvre et al. (2011) described dissolved silicate and phosphate as the most are the main limiting nutrient in the EEC."

L. 376 "Lheureux et al. (2023) documented an increase in Si:P and Si:N ratios at the SRN station in Boulogne-sur-Mer, while the SOMLIT (National Observation Service of the Research Infrastructure ILICO) coastal station (South of Boulogne-sur-Mer and further offshore) showed a decrease…"

**4-** A minor typo L. 443 "surveyx" to be corrected.